# E-Commerce Technologies Adoption Strategy Selection in Indonesian SMEs Using the Decision-Makers, Technological, Organizational and Environmental (DTOE) Framework

Saffira Annisa Bening [1], Muhammad Dachyar [1,*], Novandra Rhezza Pratama [1], Jaehyun Park [2] and Younghoon Chang [3]

1    Department of Industrial Engineering, Universitas Indonesia, Depok 16424, Indonesia
2    School of Design, Hong Kong Polytechnic University, Hong Kong 999077, China
3    School of Management and Economics, Beijing Institute of Technology, Beijing 100081, China
*    Correspondence: mdachyar08@gmail.com

**Abstract:** Small and Medium-sized Enterprises (SMEs) are key contributors to Indonesia's economy. E-commerce can assist SMEs in gaining a competitive edge and is expected to be the largest contributor to the country's digital economy by 2030. However, only 22% of SMEs in Indonesia have adopted E-commerce. Hence this study aims to investigate the criteria that influence E-commerce adoption by Indonesian retail SMEs and select the best strategy using the Decision-Makers, Technological, Organizational, and Environmental (DTOE) Framework and the Diffusion of Innovation (DOI) theory. This study aims to fill the research gap by examining the essential factors in E-commerce adoption by Indonesian SMEs in the retail industry and their key strategy to increase utilization. The influence and priorities between criteria were calculated using the Decision-Making Trial and Evaluation Laboratory (DEMATEL) based Analytic Network Process (ANP) method. Furthermore, an E-commerce adoption strategy was selected using Complex Proportional Assessment (COPRAS) method. This research showed critical criteria for adopting E-commerce technology in retail SMEs, Decision Maker's IT Knowledge, Innovativeness, and Complexity. In addition, developing or training Information Technology (IT) and E-commerce skills were identified as the best strategy to increase E-commerce adoption. This study can help raise the understanding of policymakers, E-commerce service providers, and retail SME decision-makers on influencing criteria in adopting the best strategy to be applied to increase this technology adoption.

**Keywords:** E-commerce; adoption; SMEs; multi-criteria decision making (MCDM); DEMATEL-based ANP; COPRAS

## 1. Introduction

The economic sector was severely affected by the COVID-19 pandemic. In the second quarter of 2020, its growth contracted due to a decrease in various aspects, such as household consumption and the realization of investment and government spending [1]. This led to a decline of relatively 3.56% in the third quarter of 2021. Unfortunately, this drop was caused by the slackening domestic demand in accordance with the mobility restriction policy implemented to combat the COVID-19 delta variant [2].

Strengthening Small and Medium-sized Enterprises (SMEs) is the key to revitalizing the economy because these are the main driving force [3]. Moreover, they also play a crucial role in absorbing labor, reducing unemployment, and contributing to the Gross Domestic Product (GDP). SMEs absorb as much as 97% of the workforce, contributing approximately 61.07% or 572.5 billion USD as converted from Indonesian Rupiah to the GDP [4].

SMEs must be receptive to adopting new technologies, especially in employing various digital solutions that aid in promotion while reducing production costs to restore and accelerate economic growth. Digitalization was also identified as one of the problems

associated with its growth. Therefore, it was adopted as a strategy in the National Medium-Term Development Plan for 2020 to 2024 [5]. The government's support of the digital transformation of SMEs was also proven by implementing the National Economic Recovery Program and the Proudly Made in Indonesia National Movement. These were also targeted at the digital transformation of 30 million SMEs by 2023. This number is quite ambitious considering, based on data until mid-August 2021. It was reported that relatively 14 million, or 22% of the total SMEs in the country, had adopted Ecommerce [6].

In 2020, the digital economy only contributed 4% to Indonesia's gross domestic product (GDP). However, it was anticipated to increase at least eight times, contributing approximately 18% to the GDP, thereby increasing from 42.4 billion USD to 302.57 billion USD as converted from the Indonesian Rupiah by 2030. E-commerce plays a significant role in the digital economy, equivalent to 34% or 126.88 billion USD [7]. Given that there are relatively 202 million internet users in the country, its expansion triggers a lot of potentials. In accordance with this 73% of the total 274 million population has been able to penetrate the internet [8]. The technological revolution has exposed SMEs to diverse opportunities. Based on the research carried out by [9,10], the adoption of E-commerce triggers competitive advantage and significantly impacts SME operations. Furthermore, competitive advantage is defined as quality improvement, differentiation, growth, and cost reduction [11–13].

1.  Several recent studies focus on how E-commerce is generally adopted throughout diverse sectors. As a result, there is only a little research on its utilization in certain industries, such as retail companies [14]. In Indonesia, the distribution of SMEs is dominated by the retail, repair, and vehicle maintenance industries which constitute 46% of all businesses in the country [15]. The retail company is also the second-largest contributor to the GDP after the processing industry [16]. As one of the largest business sectors that constitute SMEs, it played a vital role in recovering due to the pandemic and driving the national economy. There is a need to further investigation to ascertain how SMEs are responding to E-commerce adoption to maintain a competitive edge during the digitization era. Due to several reasons, namely, limited research surrounding E-commerce adoption in retail industries, retail being the biggest segment in Indonesian SMEs, the low adoption rate of E-commerce in Indonesian SMEs, and government support and target in the digital transformation of SMEs, this research is conducted to answer several questions: Understanding and ranking the factors that influence the decisions of Indonesian retail SMEs in the use of E-commerce technology.

2.  Develop strategic recommendations to increase E-commerce technology adoption among SMEs, especially in the retail industry.

This work is different from previous research in three aspects. First, despite numerous studies, only a few research evaluate adoption criteria that pertain to decision-makers [17,18] without simultaneously considering technological, organizational, and environmental factors. We evaluated all criteria using Decision-Makers, Technological, Organizational, and Environmental (DTOE) framework. Second, most literature tries to explain E-commerce adoption patterns without suggesting ways to boost adoption [14,19–22]. Finally, although prior literature [23] offered proposals to increase E-commerce adoption, but failed to consider which techniques should be prioritized as the primary strategy for boosting E-commerce adoption. This research is motivated by addressing current research gaps to evaluate the influence of criteria related to the decision-maker, technological, organizational, and environmental dimensions aligned with E-commerce adoption in the retail SME sector, propose alternative strategies to boost E-commerce adoption and rank alternative strategies required to increase E-commerce adoption. The influence and priorities between criteria were calculated using the Decision-Making Trial and Evaluation Laboratory (DEMATEL) based Analytic Network Process (ANP) method. Furthermore, an E-commerce adoption strategy was selected using Complex Proportional Assessment (COPRAS) method.

## 2. Theoretical Review

### 2.1. SMEs and E-Commerce

SMEs contribute significantly to the economic performance of countries around the world. It is, therefore, an essential constituent of national economic growth. These contributions include creating new job opportunities, including their enormous impact on the country's GDP [24]. The dynamic market developments have urged SMEs to adopt E-commerce to maintain their strategic competitiveness [25]. The rapid emergence and growth of digital and data technologies have triggered competition in many industries [26]. E-commerce technology strongly appeals to various companies, especially in the retail sector. Many traditional brick-and-mortar stores are becoming modernized and shifting to digital business models as E-commerce gains attention in the retail industry [27]. To remain competitive, they tend to develop online enterprises through web-based stores and mobile applications. This aids in increasing the number of digital contact points with customers and blends the offline and online worlds [28].

### 2.2. Underlying Theories

Many academics have given the study of E-commerce adoption by SMEs considerable attention to the potential advantages this technology offers businesses. Table 1 reviews the studies conducted in various countries based on SMEs' adoption of E-commerce. In addition, limited studies have been conducted to determine the criteria related to the adoption dimension [22,23], although without considering the technological, organizational, and environmental yardsticks simultaneously. Furthermore, most of these studies explain E-commerce adoption behavior without offering strategic recommendations to boost its increase. Unfortunately, such research [23] failed to assess the prioritized or the main strategies needed to increase E-commerce adoption. Due to these gaps, this research seeks to evaluate the influence of the decision-makers' criteria, including technological, organizational, and environmental dimensions aligned with E-commerce adoption in the retail SME sector, as well as rank the alternative strategies needed to boost its increase using DEMATEL based ANP (DANP) and COPRAS methods.

**Table 1.** Examples of related empirical research on SMEs' use of E-commerce.

| | Researched Aspects | | | | | |
|---|---|---|---|---|---|---|
| Source | Technology | Organization | Environment | Decision-Makers | Adoption Strategy | Framework and Theory |
| Kurnia et al. [14] | Yes | Yes | Yes | No | No | DOI, National Institutional Perspective |
| Grandon and Pearson [19] | Yes | Yes | Yes | No | No | DOI and Technology Acceptance Model |
| Abou-Shouk et al. [20] | Yes | No | No | No | No | - |
| Ghobakhloo and Tang [18] | Yes | No | No | Yes | No | DOI |
| Rana et al. [21] | Yes | Yes | Yes | No | No | - |
| Seyal et al. [23] | Yes | Yes | Yes | No | Yes | TOE |
| Gu [17] | Yes | No | No | Yes | No | DOI |
| Saffu et al. [22] | Yes | Yes | Yes | No | No | Technology Acceptance Model |
| Kendall et al. [29] | Yes | No | No | No | No | DOI |

According to Table 1, the researchers used the Diffusion of Innovation (DOI) theory proposed by Rogers [30] as their main theoretical framework to analyze E-commerce adoption in SMEs across various nations. However, the use of technology adoption theories like DOI alone does not give a detailed explanation of such complex phenomena. Instead, this makes it difficult to understand and select E-commerce adoption criteria relevant to the organizational and environmental contexts.

Other theoretical frameworks like Technology, Organization, and Environment (TOE) were criticized for not paying attention to criteria related to the managers' characteristic SMEs having centralized organizational structures, with owners making most of the crucial decisions [31]. Responding to this criticism, Thong [32] proposed that an additional component verifies the findings of the fourth dimension classified as a characteristic of CEOs or decision-makers. It is an extension of the TOE called the Decision-Makers, Technological, Organizational, and Environmental (DTOE) framework. This model tends to alleviate some of the worries that bother these SMEs regarding a highly centralized structure in which the owners make the most important decisions [31]. Because this study aims to analyze the adoption of E-commerce by SMEs, the TOE extension, namely the DTOE framework, was utilized to support this analysis.

Furthermore, this aligns with the Diffusion of Innovation (DOI) theory proposed by Rogers [30]. The integration of the DOI theory and the DTOE framework is also driven by the fact that this combination highlights individual characteristics and internal and external organizations. This study employed these two attributes because it covers all criteria that affect E-commerce use at the individual, organizational, and environmental levels, as indicated in the literature review.

### 2.3. Diffusion of Innovation (DOI) Theory

Rogers [30] proposed the Diffusion of Innovation (DOI) theory to analyze how, when, and to what extent the people and the business sector accept new ideas and technology. This concept is "the process by which innovation is communicated through channels over time among members of a social system". Innovation is defined as "an idea, practice, or object perceived as new by an individual or other unit of adoption". It aids in identifying the factors that affect how quickly a company adopts technical breakthroughs. Rogers [30] used five qualities or attributes to measure innovation. These include relative advantage, complexity, compatibility, trialability, and observability.

### 2.4. Decision-Makers, Technological, Organizational and Environmental (DTOE) Framework

Thong [32] employed the Technology, Organization, and Environment (TOE) theory proposed by Tornatzky and Fleischer [33] in four dimensions when examining the SME industry. Based on the extension of the TOE theory, Thong [32] contends that the CEO or manager makes the majority of the crucial choices in SMEs due to their highly centered organizational structure. In addition to technology, organization, and environment, Thong [32] established the significance of the fourth dimension, the CEO or decision-maker dimension. Given owners' and managers' significant role in SMEs, this study adopted Thong's [32] Decision-Makers, Technological, Organizational, and Environmental (DTOE) Framework.

## 3. Materials and Methods

### 3.1. Methodology

This research employed a combination of the DEMATEL-based Analytic Network Process (DANP) and the Complex Proportional Assessment (COPRAS) methods to determine the best alternative strategy to increase E-commerce adoption among SMEs. First, DANP was utilized due to its interdependence. It was also used to build Influential Network Relations Maps (INRM) and generate influential weights based on basic concepts from the Analytic Network Process (ANP) method. Second, the COPRAS approach with the input of influential weights from DANP was applied to determine the ranking of alternative strategies needed to boost E-commerce adoption among SMEs. The overall process is shown in Figure 1.

### 3.2. Participants and Questionnaire

Based on the previous literature study, the Decision-Makers, Technological, Organizational, and Environmental (DTOE) framework was selected to classify the investigated criteria. The DOI theory supports the technological criteria, while the others are based on

literature studies. The selected criteria were considered by different research and journals, as shown in Table 2.

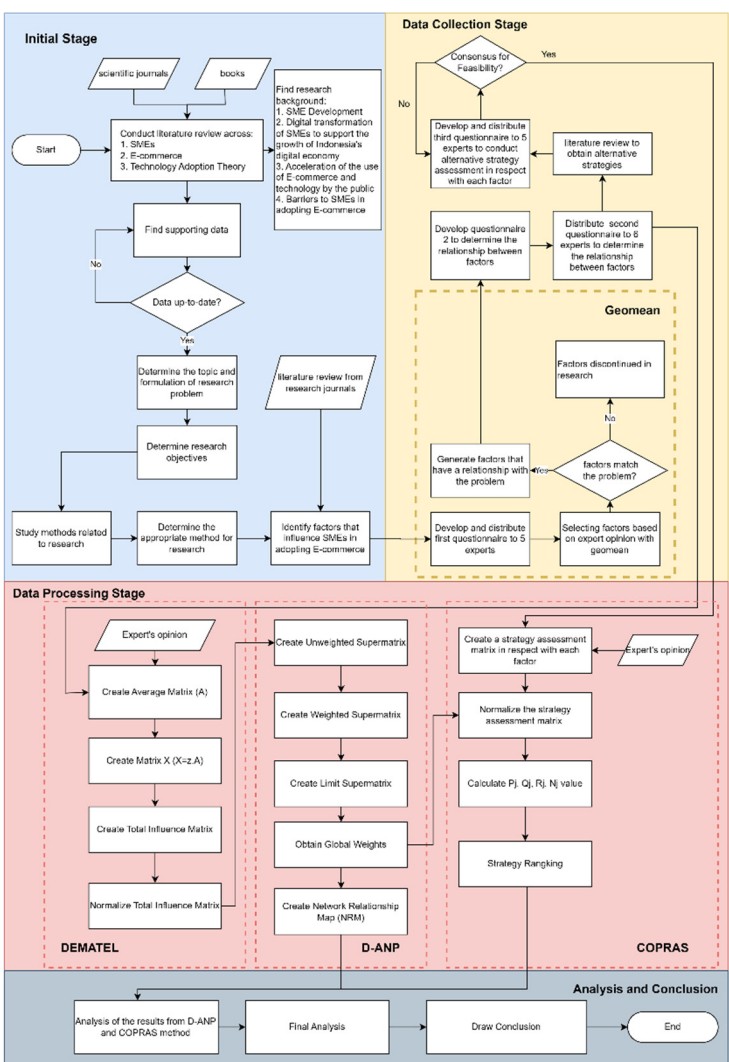

**Figure 1.** Research Methodology.

Meanwhile, three different questionnaires were utilized for data collection. These questionnaires were distributed between January and March 2022 through an online survey from experts based in Jakarta, Indonesia. These experts were selected as they come from SME and digital economy researchers, SME E-commerce business leaders, and retail SME industry players to get a holistic response between academics, business, and industry players.

The first questionnaire was distributed to five participants to ascertain the suitability of the literature study's criteria and determine the circumstances surrounding E-commerce adoption by retail SMEs in the country. These individuals are experts in the SME field and E-commerce service providers related to the retail industry with a minimum of five years of work experience. These experts were relevant to this research because it consists of business leaders ranging from managers and directors and their extensive background in E-commerce, specifically in the retail SME sector, and researchers with more than 20 years of experience.

The second questionnaire was distributed to six participants with at least five years of experience to determine influential relationships and weigh the criteria. These are business owners and E-commerce service providers for retail SMEs that have not yet adopted this

technology. The questionnaire employed pairwise comparison to assess the impact and influence of the criteria and their weight.

The third questionnaire, used to assess the best E-commerce adoption strategy, was distributed to five experts in the SME industry and digital economy as well as business owners and E-commerce service providers with a minimum of ten years of work experience. It uses a five-point scale from one to five, denoting very poor to very high performances. Coincidentally, this research is expected to assist policymakers and retail SME decision-makers identify the essential criteria for adopting E-commerce, including the most effective alternative strategy for promoting this technology.

### 3.3. Integrated DEMATEL-Based ANP (DANP)

MCDM methods can generally be divided into two types according to their compensatory or non-compensating nature, in which compensatory methods (ex. TOPSIS) usually combine performance which is categorized into functions to be optimized [34]. However, TOPSIS has some main drawbacks, including the correlation between criteria is not considered in evaluating the Euclidean distance in TOPSIS and the ambiguity of using only objective or subjective methods to determine weights [35]. Hence this method is not suitable for analyzing criteria that have interdependencies. Non-compensated methods are also known as outranking (ex., ELECTRE and PROMETHEE). The outranking approach establishes a preference relation on a set of alternatives that indicates the degree of dominance among them [36]. However, an outranking approach with a non-compensating nature cannot always offer complete ranking results [37]. In the meantime, pairwise comparison in MCDM methods such as Analytic Hierarchy Proses (AHP) and Analytic Network Process (ANP) is very useful for finding the weight of different criteria and comparing alternatives concerning a subjective criterion [36].

One of the many Multi-Criteria Decision Making (MCDM) methods, namely the Analytic Network Process (ANP) introduced by [38], is often adopted in various studies. It is a complete form of the AHP method, and when the existing criteria are independent, the hierarchical link between them becomes unidirectional and cannot simplify MCDM challenges [39]. In reality, there is an existent relationship within a group of criteria. ANP resolves this problem by combining the interdependence and reciprocity between criteria and alternatives in the decision-making model [38].

**Table 2.** The initial dimensions and criteria of research.

| Dimension | Criteria | Definition | References |
|---|---|---|---|
| Decision-Makers | Decision Maker's Innovativeness | Adopting new technology faster than others in the same social environment and a manager who likes to solve problems by changing its set-up. | Al-Qirim [40]; Ghobakhloo et al. [41]; Sánchez-Torres et al. [42] |
| | Decision Maker's IT Knowledge | Managers more experienced in IT are most likely to adopt it, reducing the uncertainties and risks associated with such a decision. | Chau et al. [43]; Nair et al. [44]; Huy et al. [45] |
| Technology | Relative Advantage | Benefits of E-commerce for the internal users and the company | Abdulkarem and Hou [46]; Hamad et al. [11]; Mohtaramzadeh et al. [47] |
| | Compatibility | How well does E-commerce fit the company's current technical infrastructure, culture, values, and work practices | Abdulkarem and Hou [46]; Hamad et al. [11]; Hoang et al. [48] |
| | Complexity | The extent to which a new idea is perceived as being difficult to understand and apply | Abdulkarem and Hou [46]; Awa et al. [49]; Hamad et al. [11] |
| | Security | To secure online payment, and transactions, prevent hacking, and malware | Amornkitvikai et al. [31]; Costa and Castro [50]; Chau et al. [43] |

**Table 2.** *Cont.*

| Dimension | Criteria | Definition | References |
|---|---|---|---|
| Organization | Employee's IT Knowledge | How employees perceive E-commerce, how experienced they are, and how much formal and informal E-commerce training they have. | Chau et al. [43]; Hoang et al. [48]; Huy et al. [45] |
| | Organization Readiness | Includes resources such as money and technology that are important for approving innovative ideas | Costa and Castro [50]; Hoang et al. [48]; Lim et al. [51] |
| | Business Size | Larger SME businesses use advanced technology because they have the monetary resources. | Abdulkarem and Hou [46]; Awa et al. [49]; Hamad et al. [11] |
| Environment | Customer Pressure | This deals with the extent of pressure from customers encouraging SMEs to adopt E-commerce | Abed [52]; Abdulkarem and Hou [46] |
| | Competitive Pressure | The amount of pressure that competitors in the same industry put on others to adopt E-commerce | Abdulkarim and Hou [46]; Hamad et al. [11]; Hussain et al. [53]; Mohtaramzadeh et al. [47]; Ocloo et al. [54] |
| | Trading Partner Pressure | The amount of pressure from business partners such as suppliers mounted on SMEs to adopt E-commerce | Abdulkarem and Hou [46]; Abed [52]; Mohtaramzadeh et al. [47] |
| | Government Support | This is legislation or guidelines to protect the stakeholders' business transactions because these SMEs have limited resources, including financial and IT capabilities. These regulatory measures are aimed at securing the Internet as a transaction medium, as well as providing financial incentives for businesses to engage in electronic commerce | Hamad et al. [11]; Mohtaramzadeh et al. [47]; Hussain et al. [53] |

The Decision-Making Trial and Evaluation, Laboratory or DEMATEL method, developed by [55], analyzes structured relationships in complex systems. Its basis is centred on digraphs, which can be separated into cause-and-effect groups through a matrix [56]. It is also used to determine the critical factors of a complex system or structure with the help of an impact relationship diagram.

In the DEMATEL-based ANP method (DANP), the initial steps are continued by using Analytical Network Process (ANP) to weigh the criteria as well as determine which is the most relevant [57]. However, to determine the weight of each criterion using the conventional ANP method, the number of clusters in each column is equally divided. Each group's weight is implicitly assumed to be the same, although the degree to which one cluster affects the other varies [58]. The limitations of this traditional ANP method led to the adoption of the DANP approach because its influential weights produce outcomes based on the fundamental ideas of ANP from the total influential matrix Tc and Td. To build INRMs (Influential Network Relation Maps) for each criterion and dimension and to enhance the classic ANP method's normalization process, both DEMATEL and DANP are employed [58].

DEMATEL is a powerful tool for examining cause-and-effect interactions since it tends to display the criteria while also considering the structural model quantitatively. On the other hand, it cannot establish the weights of each criterion, thereby leading to the adoption of ANP. This procedure helps estimate and prioritize the criteria and the existent relationships when the evaluation process is diverse and complicated.

The procedure for creating an INRM involves the adoption of steps 1 to 3. Meanwhile, the influential weights are determined from the total-influential matrix using steps 4 to 7, as stated in [58–60].

1. The first step is to create an average expert opinion matrix ($A$) = $\left[A_{ij}\right]_{nxn}$, demonstrating the direct influence of criterion $i$ on $j$. Each expert assigned this rating on a 5-point Likert scale with the lowest and highest values of 0 and 4, representing "no influence" and "very high influence", respectively. The average matrix is shown in Equation (1)

$$A = \begin{bmatrix} a_{11} & a_{1j} & a_{1n} \\ a_{i1} & a_{ij} & a_{i1} \\ a_{n1} & a_{nj} & a_{nn} \end{bmatrix} \tag{1}$$

2. The second step is to generate the initial influence matrix ($X$) with $X = \left[X_{ij}\right]_{nxn}$ by using the normalization process to determine the average matrix ($A$) and also ensuring that all the main diagonal elements are 0. The matrix $X$ assists in determining the initial influence of the existing criteria, whether given or received.

3. Calculate the complete direct or indirect influence matrix. A continuous decrease of the indirect effects of problems is evident along the powers of $X$, for example, $X^2$, $X^3, \ldots, X^h$ and $\lim_{h \to \infty} X^h = [0]_{nxn}$, where $X = \left[X_{ij}\right]_{nxn}, 0 \leq X_{ij} < 1$ and $0 \leq \sum i X_{ij} \leq 1$ or $0 \leq \sum j X_{ij} \leq 1$ and at least one summation column or row, although not all, equals one. The $X$ matrix tends to be computed using Equation (2), in which all principal diagonal elements are equal to 0.

$$X = z \cdot A, where\, z = min\{ \frac{1}{max_{1\leq i \leq n \sum_{j=1}^{n} a_{ij}}}, \frac{1}{max_{1 \leq j \leq n \sum_{i=1}^{n} a_{ij}}} \}$$
$$and\, \lim_{h \to \infty} X^h = [0]_{nxn}, 0 \leq X_{ij} \leq 1 \tag{2}$$

4. Create a total influence matrix ($T$) that can be obtained using Equation (3) to explain the influence between one criterion and another, in which I represents the identity matrix.

$$T = X + X^2 + \ldots + X^h = X(I - X)^{-1}\, when\, \lim_{h \to \infty} X^h = [0]_{nxn} \tag{3}$$

Explanation

$$T = X + X^2 + \ldots + X^h = X\left(I + X + X^2 + \ldots + X^{h-1}\right)(I - X)(I - X)^{-1}$$
$$= X\left(I - X^h\right)(I - X)^{-1}$$

Then,

$$T = X(I - X)^{-1}, when\, h \to \infty$$

If vectors $r$ and $s$ are defined as the sum of rows and columns, respectively, in the total influence matrix $T$ through Equation (4), then

$$T = [t_{ij}], where\, i, j = 1,2, \ldots, n,$$

$$r = [r_i]_{n \times 1} = \left[\sum_{j=1}^{n} t_{ij}\right]_{n \times 1}, s = [s_j]_{n \times 1} = \left[\sum_{i=1}^{n} t_{ij}\right]'_{1 \times n} \tag{4}$$

If $r_i$ depicts the sum of the $i$th row in the matrix $T$, then represents the sum of direct and indirect impacts of criterion $i$ on the other criteria. If $s_j$ depicts the sum of the $j$th column in the matrix $T$, then it represents is the sum of direct and indirect effects received by criterion $j$ from the other criteria. When $j = i$ (the total number of rows and columns), $(r_i + s_j)$ represents the degree of importance of influence given and received. Furthermore, $(r_i - s_j)$ represents the net effect of criteria $i$. If $(r_i - s_j)$ has a positive value, then criterion

$i$ influences the other criteria., When $(r_i - s_j)$ has a negative value, then criterion $i$ is influenced by the other criteria.

5. The fourth step is creating an unweighted supermatrix, where the matrix prioritises the existing criteria locally: the total effect matrix $T_C = [t_{ij}]_{n \times n}$ is generated from the criteria, while $T_D = [t_{ij}^D]_{m \times m}$ is derived from the dimensions of $T_C$. In addition, the $T_C$ and $T_D$ matrices are normalized to obtain $T_C^\alpha$ and $T_D^\alpha$. The total effect matrix for both criteria and dimensions is obtained from the DEMATEL calculations. $T_D$ is the average value of the corresponding $T_C$ dimension. The process of normalizing the matrix is shown in Equation (5)

$$
T_C = \begin{array}{c}
\begin{array}{ccccccccc}
& D_1 & & \cdots & & D_j & & \cdots & & D_n & \\
c_{11} & \cdots & c_{1m_1} & & c_{j1} & \cdots & c_{jm_j} & & c_{n1} & \cdots & c_{nm_n}
\end{array}
\end{array}
\begin{array}{c}
D_1 \begin{cases} c_{11} \\ \vdots \\ c_{1m_1} \end{cases} \\
\vdots \\
D_i \begin{cases} c_{i1} \\ \vdots \\ c_{im_1} \end{cases} \\
\vdots \\
D_n \begin{cases} c_{n1} \\ \vdots \\ c_{nm_n} \end{cases}
\end{array}
\begin{bmatrix}
T_c^{11} & \cdots & T_c^{1j} & \cdots & T_c^{1n} \\
\vdots & & \vdots & & \vdots \\
T_c^{i1} & \cdots & T_c^{ij} & \cdots & T_c^{in} \\
\vdots & & \vdots & & \vdots \\
T_c^{n1} & \cdots & T_c^{nj} & \cdots & T_c^{nn}
\end{bmatrix}
\tag{5}
$$

After performing the normalization process on the total influence matrix $T_C$, the $T_C^\alpha$ with dimensions (clusters) shown in Equation (6) was established:

$$
T_c^\alpha = \begin{array}{c}
\begin{array}{ccccccccc}
& D_1 & & \cdots & & D_j & & \cdots & & D_n & \\
c_{11} & \cdots & c_{1m_1} & & c_{j1} & \cdots & c_{jm_j} & & c_{n1} & \cdots & c_{nm_n}
\end{array}
\end{array}
\begin{array}{c}
D_1 \begin{cases} c_{11} \\ \vdots \\ c_{1m_1} \end{cases} \\
\vdots \\
D_i \begin{cases} c_{i1} \\ \vdots \\ c_{im_1} \end{cases} \\
\vdots \\
D_n \begin{cases} c_{n1} \\ \vdots \\ c_{nm_n} \end{cases}
\end{array}
\begin{bmatrix}
T_c^{\alpha 11} & \cdots & T_c^{\alpha 1j} & \cdots & T_c^{\alpha 1n} \\
\vdots & & \vdots & & \vdots \\
T_c^{\alpha i1} & \cdots & T_c^{\alpha ij} & \cdots & T_c^{\alpha in} \\
\vdots & & \vdots & & \vdots \\
T_c^{\alpha n1} & \cdots & T_c^{\alpha nj} & \cdots & T_c^{\alpha nn}
\end{bmatrix}
\tag{6}
$$

The normalization of $T_C^{\alpha 11}$ is shown in Equations (7) and (8)

$$
d_{ci}^{11} = \sum_{j=1}^{m_1} t_{cij}^{11}, i = 1, 2, \ldots, m_1
\tag{7}
$$

$$
T_C^{\alpha 11} = \begin{bmatrix} t_{C11}^{11}/d_{c1}^{11} & \cdots & t_{C1j}^{11}/d_{c1}^{11} & \cdots & t_{C1m_1}^{11}/d_{c1}^{11} \\ \vdots & & \vdots & & \vdots \\ t_{Ci1}^{11}/d_i^{11} & \cdots & t_{Cij}^{11}/d_{ci}^{11} & \cdots & t_{C1m_1}^{11}/d_{ci}^{11} \\ \vdots & & \vdots & & \vdots \\ t_{Cm_11}^{11}/d_{m_1}^{11} & \cdots & t_{Cm_1j}^{11}/d_{m_1}^{11} & \cdots & t_{Cm_1m_1}^{11}/d_{cm_1}^{11} \end{bmatrix}
$$

(8)

$$
= \begin{bmatrix} t_{C11}^{\alpha 11} & \cdots & t_{C1j}^{\alpha 11} & \cdots & t_{C1m_1}^{\alpha 11} \\ \vdots & & \vdots & & \vdots \\ t_{Ci1}^{\alpha 11} & \cdots & t_{Cij}^{\alpha 11} & \cdots & t_{C1m_1}^{\alpha 11} \\ \vdots & & \vdots & & \vdots \\ t_{Cm_11}^{\alpha 11} & \cdots & t_{Cm_1j}^{\alpha 11} & \cdots & t_{Cm_1m_1}^{\alpha 11} \end{bmatrix}
$$

6. The outcome is an unweighted supermatrix, obtained by transforming the normalized total influence matrix $T_C^\alpha$ by its dimension (cluster), as shown in Equation (9)

$$
W = (T_c^\alpha)' = \begin{array}{c} \\ \\ D_1 \\ \\ \\ \\ \\ D_i \\ \\ \\ \\ D_n \\ \\ \end{array}
\begin{array}{c} \\ c_{11} \\ \vdots \\ c_{1m_1} \\ \vdots \\ c_{i1} \\ \vdots \\ c_{im_1} \\ \vdots \\ c_{n1} \\ \vdots \\ c_{nm_n} \end{array}
\begin{bmatrix} W^{11} & \cdots & W^{1j} & \cdots & W^{1n} \\ \vdots & & \vdots & & \vdots \\ W^{i1} & \cdots & W^{ij} & \cdots & W^{in} \\ \vdots & & \vdots & & \vdots \\ W^{n1} & \cdots & W^{nj} & \cdots & W^{nn} \end{bmatrix}
$$

(9)

with column headers $D_1 \; \cdots \; D_j \; \cdots \; D_n$ over $c_{11} \cdots c_{1m_1} \quad c_{j1} \cdots c_{jm_j} \quad c_{n1} \cdots c_{nm_n}$

7. Forming a weighted super matrix by normalizing the total influence matrix of dimension $T_D$ as shown in Equation (10)

$$
T_D = \begin{bmatrix} t_D^{11} & \cdots & t_D^{1j} & \cdots & t_D^{1n} \\ \vdots & & \vdots & & \vdots \\ t_D^{i1} & \cdots & t_D^{ij} & \cdots & t_D^{in} \\ \vdots & & \vdots & & \vdots \\ t_D^{n1} & \cdots & t_D^{nj} & \cdots & t_D^{nn} \end{bmatrix}
$$

(10)

To normalize the total influence matrix of dimension $T_D$, each element in the matrix $T_D$ is divided by the total of each column resulting in a new matrix $T_D^\alpha$, as proven by Equation (11) (where $t_D^{\alpha ij} = t_D^{ij}/d_i$)

$$
\begin{aligned}
T_D^\alpha &= \begin{bmatrix}
t_D^{11}/d_1 & \cdots & t_D^{1j}/d_1 & \cdots & t_D^{1n}/d_1 \\
\vdots & & \vdots & & \vdots \\
t_D^{i1}/d_i & \cdots & t_D^{ij}/d_i & \cdots & t_D^{in}/d_i \\
\vdots & & \vdots & & \vdots \\
t_D^{n1}/d_n & \cdots & t_D^{nj}/d_n & \cdots & t_D^{nn}/d_n
\end{bmatrix} \\[2mm]
&= \begin{bmatrix}
t_D^{\alpha 11} & \cdots & t_D^{\alpha 1j} & \cdots & t_D^{\alpha 1n} \\
\vdots & & \vdots & & \vdots \\
t_D^{\alpha i1} & \cdots & t_D^{\alpha ij} & \cdots & t_D^{\alpha in} \\
\vdots & & \vdots & & \vdots \\
t_D^{\alpha n1} & \cdots & t_D^{\alpha nj} & \cdots & t_D^{\alpha nn}
\end{bmatrix}
\end{aligned}
\tag{11}
$$

where $d_i = \sum_{j=1}^{n} t_D^{ij}$.

The weighted super matrix is then obtained by multiplying the normalized total influence matrix $T_D^\alpha$ with the unweighted supermatrix Was shown in Equation (12)

$$
W^\alpha = T_D^\alpha \times W = \begin{bmatrix}
t_D^{\alpha 11} \times W^{11} & \cdots & t_D^{\alpha i1} \times W^{i1} & \cdots & t_D^{\alpha 1n} \times W^{n1} \\
\vdots & & \vdots & & \vdots \\
t_D^{\alpha 1j} \times W^{1j} & \cdots & t_D^{\alpha ij} \times W^{ij} & \cdots & t_D^{\alpha nj} \times W^{nj} \\
\vdots & & \vdots & & \vdots \\
t_D^{\alpha 1n} \times W^{1n} & \cdots & t_D^{\alpha ijn} \times W^{in} & \cdots & t_D^{\alpha nn} \times W^{nn}
\end{bmatrix}
\tag{12}
$$

8.  Obtaining the limit of the weighted supermatrix by repeatedly multiplying it by its matrix until a long-term stable supermatrix is produced. This is performed to acquire the global priority vectors, also known as DANP influential weights. Moreover, the influential weight $W = (W_1, \ldots, W_j, \ldots, W_n)$ of each criterion is obtained according to the diagonal of $\lim_{g \to \infty} (W^\alpha)^g$.

### 3.4. Complex Proportional Assessment (COPRAS)

The Complex Proportional Assessment (COPRAS) approach designed by Zavadskas et al. [61] is usually applied in circumstances where a decision-maker is forced to select between numerous alternatives while considering a set of typically contradictory criteria [62]. The following are some benefits of this approach: (1) it enables the simultaneous consideration of the ratio to both the ideal and negative solutions, (2) calculations are straightforward and logical, and (3) answers are obtained faster than with other approaches such as AHP and ANP [63]. The ideal solution maximizes benefits while minimizing costs. The opposite, a negative ideal solution, maximizes costs while minimizing benefits. Chatterjee et al. [64] discovered that the correlation coefficient values between COPRAS, EVAMIX, AHP, TOPSIS, and VIKOR in the selection of materials proved that the COPRAS method has the best performance.

The procedure for applying the COPRAS method based on [62] consists of 12 steps, which are illustrated as follows:

1.  Choosing a set of criteria and alternatives,
2.  Creating a decision-making matrix X, as shown in Equation (13).

$$X = \begin{pmatrix} X_{11} & X_{12} & \ldots & X_{1m} \\ X_{21} & X_{22} & \ldots & X_{2m} \\ \vdots & \vdots & \ddots & \vdots \\ X_{n1} & X_{n2} & \ldots & X_{nm} \end{pmatrix} i = \overline{1,n} \text{ and } j = \overline{1,m} \tag{13}$$

where attribute $j$ appears in the alternative $i$ of a solution, $m$ and $n$ represent the number of attributes and evaluated possibilities, respectively.

3. Determine the significance of the criteria.

4. Conducting the normalization process on decision-making matrix $\overline{X}$ using Equation (14).

$$\overline{X}_{ij} = \frac{X_{ij}}{\sum_{j=1}^{n} X_{ij}}; i = \overline{1,n} \text{ and } j = \overline{1,m} \tag{14}$$

Equation (15) is used to obtain the result after performing the normalization process on the decision-making matrix $\overline{X}$:

$$\overline{X} = \begin{pmatrix} \overline{X}_{11} & \overline{X}_{12} & \ldots & \overline{X}_{1m} \\ \overline{X}_{21} & \overline{X}_{22} & \ldots & \overline{X}_{2m} \\ \vdots & \vdots & \ddots & \vdots \\ \overline{X}_{n1} & \overline{X}_{n2} & \ldots & \overline{X}_{nm} \end{pmatrix} \tag{15}$$

5. The weighted and normalized value decision-making matrix $\hat{X}$ is determined using Equation (16),

$$\hat{X}_{ij} = \overline{X}_{ij} \times q_j; i = \overline{1,n} \text{ and } j = \overline{1,m} \tag{16}$$

where: $q_j$ is the weight of the $i$-th criterion. The result of carrying out the normalization process on the weighted decision-making matrix is shown in Equation (17):

$$\hat{X} = \begin{pmatrix} \hat{X}_{11} & \hat{X}_{12} & \ldots & \hat{X}_{1m} \\ \hat{X}_{21} & \hat{X}_{22} & \ldots & \hat{X}_{2m} \\ \vdots & \vdots & \ddots & \vdots \\ \hat{X}_{n1} & \hat{X}_{n2} & \ldots & \hat{X}_{nm} \end{pmatrix} i = \overline{1,n} \text{ and } j = \overline{1,m} \tag{17}$$

6. The sum of $P_i$ is calculated using Equation (18), in which it is more preferred to have a larger value:

$$P_i = \sum_{j=1}^{k} \hat{X}_{ij}; \tag{18}$$

7. The sum of $R_i$ is calculated using Equation (19), in which it is more preferred to have a smaller value:

$$R_i = \sum_{j=k+1}^{m} \hat{X}_j; \tag{19}$$

Equation (19) $(m - k)$ reflects the number of criteria that must be minimized.

8. The minimal value of $R_i$ is calculated using Equation (20):

$$R_{min} = minR_i; \ i = 1, 2, \ldots, n \tag{20}$$

9. Equation (21) determines the relative significance of each alternative $Q_i$:

$$Q_i = P_i + \frac{R_{min}\sum_{i=1}^{n} R_i}{R_i\sum_{i=1}^{n} \frac{R_{min}}{R_i}} \tag{21}$$

10. Determining the rankings of each of the prioritized alternatives
11. The degree of utility of each alternative is calculated using Equation (22):

$$N_i = \frac{Q_i}{Q_{max}} \times 100\% \tag{22}$$

$Q_i$ represents the significances for each alternative derived from Equation (21) while $Q_{max}$ represents the optimal value for each alternative $Q_i$.

## 4. Results

### 4.1. Analyzing the Connections between Dimensions and Criteria to Create an INRM

Based on the result of the survey obtained from the first questionnaire, an assessment of each criterion was performed by five experts using the geomean value with a five-point Likert scale. Meanwhile, a geomean value greater than 3.5 was accepted, resulting in 11 of 13 criteria selections, as shown in Table 3. These dimensions are based on the extension of the TOE theory, in which the CEO or manager makes most of the crucial choices in SMEs due to their highly centered organizational structure; hence DTOE framework was applied. To support this framework, the Diffusion of Innovation (DOI) theory to analyze how, when, and to what extent the people and the business sector accept new ideas and technology that include relative advantage, complexity, and compatibility is integrated into this research.

**Table 3.** Selected dimensions and criteria of research.

| Dimension | Criteria |
|---|---|
| Decision-Makers (D1) | Decision Maker's Innovativeness (C1)<br>Decision Maker's IT Knowledge (C2) |
| Technology (D2) | Relative Advantage (C3)<br>Compatibility (C4) |
| | Complexity (C5)<br>Security (C6) |
| Organization (D3) | Employee's IT Knowledge (C7)<br>Organization Readiness (C8) |
| Environment (D4) | Customer Pressure (C9)<br>Competitive Pressure (C10)<br>Government Support (C11) |

The DEMATEL method was then used to analyze 11 criteria within four dimensions evaluated by six experts. First, the average matrix (*A*) was obtained from pairwise assessments of the requirements regarding influences and directions using Equation (1), as shown in Table 4. Afterwards, the normalized direct-influence matrix (*X*) was constructed using Equation (2), as shown in Table 5. Next, Equation (3) was used to calculate the inverse of the difference between matrix *X* (Table 5) and identity matrix *I*. This was multiplied by Matrix *X*, and the total influence matrix for criteria ($T_C$) was realized, as shown in Table 6. Finally, the average criteria within the relevant dimensions are realized to obtain the total influence matrix for dimension ($T_D$), as shown in Table 7.

**Table 4.** The average matrix (*A*).

|     | C1   | C2    | C3    | C4    | C5    | C6    | C7    | C8    | C9    | C10   | C11   | Total |
|-----|------|-------|-------|-------|-------|-------|-------|-------|-------|-------|-------|-------|
| C1  | 0    | 3.33  | 3.50  | 3.50  | 2.67  | 3.33  | 3.00  | 3.17  | 2.83  | 3.50  | 2.83  | 31.67 |
| C2  | 3.33 | 0     | 3.50  | 3.50  | 3.00  | 3.50  | 3.00  | 3.00  | 3.17  | 3.50  | 3.17  | 32.67 |
| C3  | 3.50 | 3.17  | 0     | 3.33  | 3.00  | 3.50  | 3.17  | 3.00  | 3.00  | 3.50  | 3.00  | 32.17 |
| C4  | 3.33 | 3.17  | 3.00  | 0     | 3.17  | 3.17  | 3.17  | 3.33  | 3.00  | 3.50  | 3.17  | 32.00 |
| C5  | 3.33 | 3.33  | 3.17  | 3.33  | 0     | 3.50  | 3.33  | 3.33  | 3.33  | 3.67  | 3.33  | 33.67 |
| C6  | 3.67 | 3.33  | 3.33  | 3.50  | 3.17  | 0     | 3.50  | 3.33  | 3.50  | 3.50  | 3.17  | 34.00 |
| C7  | 3.33 | 3.17  | 3.00  | 2.83  | 3.17  | 3.67  | 0     | 3.17  | 2.67  | 2.67  | 2.83  | 30.50 |
| C8  | 3.00 | 2.50  | 2.67  | 2.83  | 2.83  | 3.00  | 2.67  | 0     | 2.67  | 2.67  | 2.50  | 27.33 |
| C9  | 2.83 | 2.50  | 2.50  | 2.67  | 2.33  | 3.17  | 2.33  | 2.33  | 0     | 2.50  | 2.17  | 25.33 |
| C10 | 2.83 | 2.67  | 2.83  | 2.83  | 2.83  | 2.67  | 2.33  | 2.50  | 3.00  | 0     | 2.67  | 27.17 |
| C11 | 2.83 | 3.00  | 2.67  | 3.17  | 2.67  | 3.17  | 2.83  | 3.00  | 2.67  | 2.50  | 0     | 28.50 |
| Total | 32 | 30.17 | 30.17 | 31.50 | 28.83 | 32.67 | 29.33 | 30.17 | 29.83 | 31.50 | 28.83 |       |

**Table 5.** The normalized direct-influence matrix (*X*) for criteria.

|     | C1   | C2   | C3   | C4   | C5   | C6   | C7   | C8   | C9   | C10  | C11  | Total |
|-----|------|------|------|------|------|------|------|------|------|------|------|-------|
| C1  | 0.00 | 0.10 | 0.10 | 0.10 | 0.08 | 0.10 | 0.09 | 0.09 | 0.08 | 0.10 | 0.08 | 0.93  |
| C2  | 0.10 | 0.00 | 0.10 | 0.10 | 0.09 | 0.10 | 0.09 | 0.09 | 0.09 | 0.10 | 0.09 | 0.96  |
| C3  | 0.10 | 0.09 | 0.00 | 0.10 | 0.09 | 0.10 | 0.09 | 0.09 | 0.09 | 0.10 | 0.09 | 0.95  |
| C4  | 0.10 | 0.09 | 0.09 | 0.00 | 0.09 | 0.09 | 0.09 | 0.10 | 0.09 | 0.10 | 0.09 | 0.94  |
| C5  | 0.10 | 0.10 | 0.09 | 0.10 | 0.00 | 0.10 | 0.10 | 0.10 | 0.10 | 0.11 | 0.10 | 0.99  |
| C6  | 0.11 | 0.10 | 0.10 | 0.10 | 0.09 | 0.00 | 0.10 | 0.10 | 0.10 | 0.10 | 0.09 | 1.00  |
| C7  | 0.10 | 0.09 | 0.09 | 0.08 | 0.09 | 0.11 | 0.00 | 0.09 | 0.08 | 0.08 | 0.08 | 0.90  |
| C8  | 0.09 | 0.07 | 0.08 | 0.08 | 0.08 | 0.09 | 0.08 | 0.00 | 0.08 | 0.08 | 0.07 | 0.80  |
| C9  | 0.08 | 0.07 | 0.07 | 0.08 | 0.07 | 0.09 | 0.07 | 0.07 | 0.00 | 0.07 | 0.06 | 0.75  |
| C10 | 0.08 | 0.08 | 0.08 | 0.08 | 0.08 | 0.08 | 0.07 | 0.07 | 0.09 | 0.00 | 0.08 | 0.80  |
| C11 | 0.08 | 0.09 | 0.08 | 0.09 | 0.08 | 0.09 | 0.08 | 0.09 | 0.08 | 0.07 | 0.00 | 0.84  |
| Total | 0.94 | 0.89 | 0.89 | 0.93 | 0.85 | 0.96 | 0.86 | 0.89 | 0.88 | 0.93 | 0.85 | 9.85 |

**Table 6.** The total influence matrix for criteria (*T*<sub>C</sub>).

|     | C1   | C2   | C3   | C4   | C5   | C6   | C7   | C8   | C9   | C10  | C11  | Total |
|-----|------|------|------|------|------|------|------|------|------|------|------|-------|
| C1  | 0.77 | 0.82 | 0.83 | 0.85 | 0.77 | 0.87 | 0.79 | 0.82 | 0.80 | 0.86 | 0.78 | 8.97  |
| C2  | 0.88 | 0.75 | 0.85 | 0.88 | 0.80 | 0.90 | 0.81 | 0.83 | 0.83 | 0.88 | 0.81 | 9.22  |
| C3  | 0.88 | 0.83 | 0.74 | 0.86 | 0.79 | 0.89 | 0.81 | 0.82 | 0.82 | 0.87 | 0.79 | 9.10  |
| C4  | 0.87 | 0.82 | 0.82 | 0.77 | 0.79 | 0.88 | 0.80 | 0.83 | 0.81 | 0.86 | 0.79 | 9.04  |
| C5  | 0.90 | 0.86 | 0.86 | 0.89 | 0.74 | 0.92 | 0.84 | 0.86 | 0.85 | 0.90 | 0.83 | 9.46  |
| C6  | 0.92 | 0.87 | 0.87 | 0.90 | 0.83 | 0.84 | 0.85 | 0.87 | 0.86 | 0.90 | 0.83 | 9.55  |
| C7  | 0.84 | 0.79 | 0.79 | 0.82 | 0.77 | 0.86 | 0.69 | 0.79 | 0.78 | 0.81 | 0.76 | 8.69  |
| C8  | 0.76 | 0.71 | 0.71 | 0.74 | 0.69 | 0.77 | 0.70 | 0.64 | 0.71 | 0.74 | 0.68 | 7.84  |
| C9  | 0.71 | 0.67 | 0.67 | 0.69 | 0.64 | 0.73 | 0.65 | 0.66 | 0.59 | 0.69 | 0.63 | 7.31  |
| C10 | 0.75 | 0.71 | 0.71 | 0.74 | 0.69 | 0.75 | 0.68 | 0.70 | 0.71 | 0.66 | 0.68 | 7.78  |
| C11 | 0.78 | 0.75 | 0.74 | 0.78 | 0.71 | 0.80 | 0.72 | 0.75 | 0.73 | 0.76 | 0.64 | 8.15  |
| Total | 9.05 | 8.57 | 8.58 | 8.92 | 8.22 | 9.20 | 8.35 | 8.57 | 8.49 | 8.93 | 8.22 |     |

**Table 7.** The total influence matrix for dimension (*T*<sub>D</sub>).

|         | D1   | D2   | D3   | D4   | Total (*r*) |
|---------|------|------|------|------|-------------|
| D1      | 0.81 | 0.84 | 0.81 | 0.82 | 3.29        |
| D2      | 0.87 | 0.84 | 0.84 | 0.84 | 3.38        |
| D3      | 0.77 | 0.77 | 0.71 | 0.75 | 2.99        |
| D4      | 0.72 | 0.72 | 0.69 | 0.68 | 2.81        |
| Total (*s*) | 3.17 | 3.17 | 3.05 | 3.09 |         |

The value of $r$ in the dimensions is obtained by adding all the elements in the $(T_D)$ rows. Meanwhile, the $s$ value is realized by adding all the elements in the $(T_D)$ columns, as seen in Equation (4). Table 8 summarizes the dimensions $r$ and $s$ values or received and given influences.

**Table 8.** The total of dimensions received and given influences.

| Dimension | $r$ | $s$ | $r + s$ | $r - s$ |
|---|---|---|---|---|
| Decision-Makers (D1) | 3.29 | 3.17 | 6.46 | 0.12 |
| Technology (D2) | 3.38 | 3.17 | 6.55 | 0.22 |
| Organization (D3) | 2.99 | 3.05 | 6.04 | −0.06 |
| Environment (D4) | 2.81 | 3.09 | 5.91 | −0.28 |

The calculation of the $r$ and $s$ values for the criteria is slightly different compared to the dimensions. The R-value is obtained in the criteria by adding all the row elements in the related dimension. However, the $s$ value is obtained by adding all the column elements in the related dimension. The summary of both values or received and given influences are shown in Table 9.

**Table 9.** The total of criteria received and given influences.

| Criterion | $r$ | $s$ | $r + s$ | $r - s$ |
|---|---|---|---|---|
| Decision Maker's Innovativeness (C1) | 1.59 | 1.66 | 3.25 | −0.06 |
| Decision Maker's IT Knowledge (C2) | 1.64 | 1.57 | 3.21 | 0.06 |
| Relative Advantage (C3) | 3.29 | 3.29 | 6.58 | 0.00 |
| Compatibility (C4) | 3.25 | 3.42 | 6.68 | −0.17 |
| Complexity (C5) | 3.41 | 3.16 | 6.57 | 0.25 |
| Security (C6) | 3.44 | 3.52 | 6.97 | −0.08 |
| Employee's IT Knowledge (C7) | 1.49 | 1.39 | 2.87 | 0.10 |
| Organization Readiness (C8) | 1.33 | 1.43 | 2.77 | −0.10 |
| Customer Pressure (C9) | 1.91 | 2.03 | 3.94 | −0.12 |
| Competitive Pressure (C10) | 2.05 | 2.11 | 4.16 | −0.06 |
| Government Support (C11) | 2.13 | 1.95 | 4.08 | 0.18 |

INRM was generated using the $r$ and $s$ values from the total dimensions received and given influences (Table 8), as shown in Figure 2.

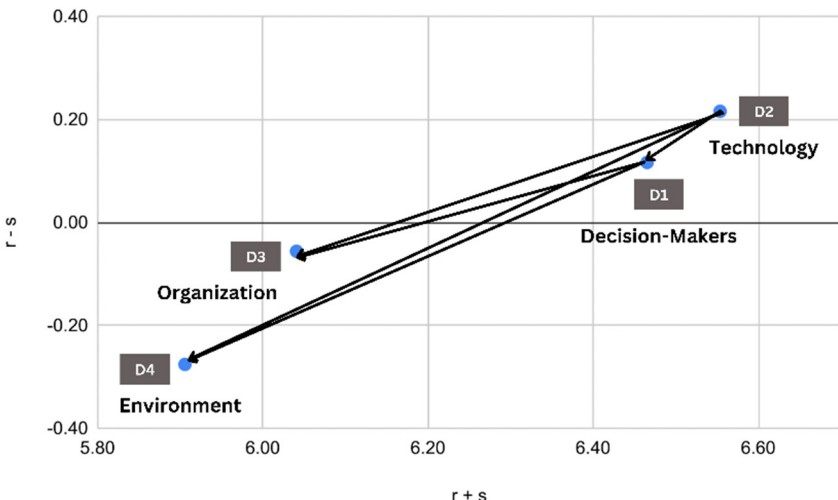

**Figure 2.** The INRM between dimensions.

INRM within four dimensions was generated using the $r$ and $s$ values from the total criteria received and given influences (Table 9), as shown in Figure 3.

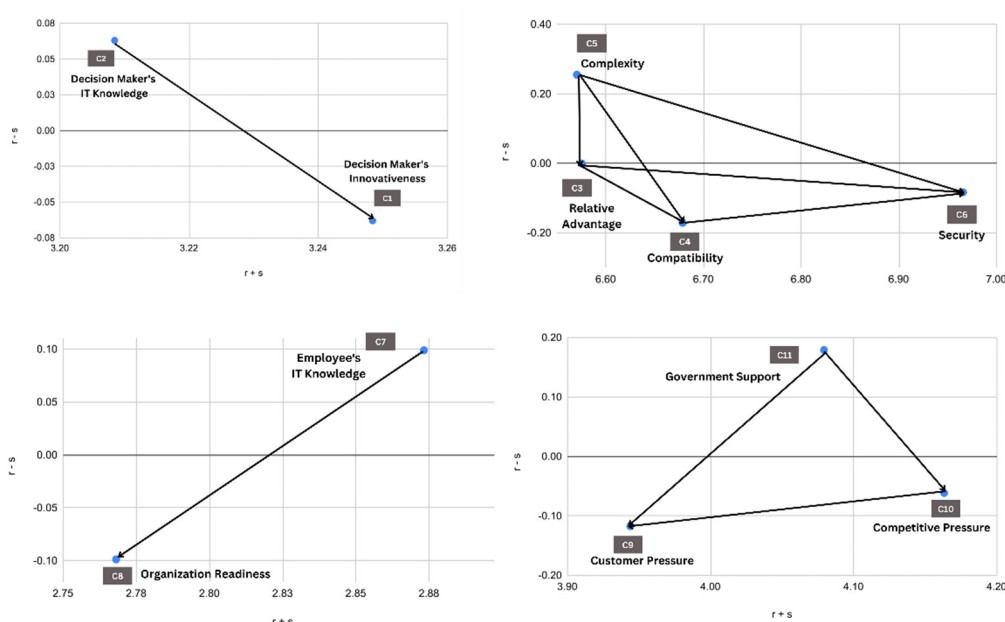

**Figure 3.** The INRM is within four dimensions.

### 4.2. Influential Weights for E-commerce Adoption in Retail SMEs

The influential weights were calculated, and the relationship structure of the SMEs' E-commerce adoption criteria was established. ($T_C$) was then normalized using Equations (5)–(8) ($T_C^\alpha$) and transposed by Equation (9) to calculate an unweighted supermatrix ($W_C$) = $(T_C^\alpha)'$ as shown in Table 10. Furthermore, an unweighted supermatrix dimension ($W_D$) was obtained by transposing the normalized matrix of $T_D$ ($T_D^\alpha$), resulting in $W_D = (T_D^\alpha)'$ as shown in Table 11 using the same equations as in for criteria. The weighted supermatrix is calculated by multiplying $W_C$ and $W_D$ as seen in Equations (10)–(12), and the results are shown in Table 12. The limit of the weighted supermatrix's power was calculated until it reached a steady state $\lim_{g\to\infty} (W^\alpha)^g$ as shown in Table 13. The global weight of the criteria was obtained using the diagonal value of the limit supermatrix. The local weight of each criterion was then computed by dividing the global weight by its total in the same dimension. Table 14 shows the global and local weights for each criterion and dimension.

**Table 10.** Unweighted Supermatrix of Criteria ($W_C$).

|      | C1   | C2   | C3   | C4   | C5   | C6   | C7   | C8   | C9   | C10  | C11  |
|------|------|------|------|------|------|------|------|------|------|------|------|
| C1   | 0.48 | 0.54 | 0.51 | 0.51 | 0.51 | 0.51 | 0.51 | 0.52 | 0.52 | 0.51 | 0.51 |
| C2   | 0.52 | 0.46 | 0.49 | 0.49 | 0.49 | 0.49 | 0.49 | 0.48 | 0.48 | 0.49 | 0.49 |
| C3   | 0.25 | 0.25 | 0.23 | 0.25 | 0.25 | 0.25 | 0.24 | 0.24 | 0.24 | 0.25 | 0.24 |
| C4   | 0.26 | 0.26 | 0.26 | 0.24 | 0.26 | 0.26 | 0.25 | 0.25 | 0.25 | 0.26 | 0.26 |
| C5   | 0.23 | 0.23 | 0.24 | 0.24 | 0.22 | 0.24 | 0.24 | 0.24 | 0.23 | 0.24 | 0.23 |
| C6   | 0.26 | 0.26 | 0.27 | 0.27 | 0.27 | 0.24 | 0.27 | 0.26 | 0.27 | 0.26 | 0.26 |
| C7   | 0.49 | 0.49 | 0.50 | 0.49 | 0.49 | 0.50 | 0.47 | 0.52 | 0.49 | 0.49 | 0.49 |
| C8   | 0.51 | 0.51 | 0.50 | 0.51 | 0.51 | 0.50 | 0.53 | 0.48 | 0.51 | 0.51 | 0.51 |
| C9   | 0.33 | 0.33 | 0.33 | 0.33 | 0.33 | 0.33 | 0.33 | 0.33 | 0.31 | 0.35 | 0.34 |
| C10  | 0.35 | 0.35 | 0.35 | 0.35 | 0.35 | 0.35 | 0.35 | 0.35 | 0.36 | 0.32 | 0.36 |
| C11  | 0.32 | 0.32 | 0.32 | 0.32 | 0.32 | 0.32 | 0.32 | 0.32 | 0.33 | 0.33 | 0.30 |

**Table 11.** Unweighted Supermatrix of Dimension ($W_D$).

|  | D1 | D2 | D3 | D4 | Average |
|---|---|---|---|---|---|
| D1 | 0.25 | 0.26 | 0.26 | 0.26 | 0.25 |
| D2 | 0.26 | 0.25 | 0.26 | 0.26 | 0.25 |
| D3 | 0.25 | 0.25 | 0.24 | 0.25 | 0.24 |
| D4 | 0.25 | 0.25 | 0.25 | 0.24 | 0.25 |
| Total | 1.00 | 1.00 | 1.00 | 1.00 |  |

**Table 12.** Weighted Supermatrix.

|  | C1 | C2 | C3 | C4 | C5 | C6 | C7 | C8 | C9 | C10 | C11 |
|---|---|---|---|---|---|---|---|---|---|---|---|
| C1 | 0.12 | 0.13 | 0.13 | 0.13 | 0.13 | 0.13 | 0.13 | 0.13 | 0.13 | 0.13 | 0.13 |
| C2 | 0.13 | 0.11 | 0.12 | 0.12 | 0.13 | 0.12 | 0.13 | 0.13 | 0.12 | 0.13 | 0.13 |
| C3 | 0.06 | 0.06 | 0.06 | 0.06 | 0.06 | 0.06 | 0.06 | 0.06 | 0.06 | 0.06 | 0.06 |
| C4 | 0.07 | 0.07 | 0.06 | 0.06 | 0.06 | 0.06 | 0.06 | 0.06 | 0.07 | 0.07 | 0.07 |
| C5 | 0.06 | 0.06 | 0.06 | 0.06 | 0.05 | 0.06 | 0.06 | 0.06 | 0.06 | 0.06 | 0.06 |
| C6 | 0.07 | 0.07 | 0.07 | 0.07 | 0.07 | 0.06 | 0.07 | 0.07 | 0.07 | 0.07 | 0.07 |
| C7 | 0.12 | 0.12 | 0.12 | 0.12 | 0.12 | 0.12 | 0.11 | 0.12 | 0.12 | 0.12 | 0.12 |
| C8 | 0.13 | 0.13 | 0.12 | 0.13 | 0.12 | 0.12 | 0.13 | 0.11 | 0.12 | 0.12 | 0.12 |
| C9 | 0.08 | 0.08 | 0.08 | 0.08 | 0.08 | 0.08 | 0.08 | 0.08 | 0.07 | 0.08 | 0.08 |
| C10 | 0.09 | 0.09 | 0.09 | 0.09 | 0.09 | 0.09 | 0.09 | 0.09 | 0.09 | 0.08 | 0.09 |
| C11 | 0.08 | 0.08 | 0.08 | 0.08 | 0.08 | 0.08 | 0.08 | 0.08 | 0.08 | 0.08 | 0.07 |

**Table 13.** The limit supermatrix when $\lim_{g \to \infty} (W^\alpha)^g$.

|  | C1 | C2 | C3 | C4 | C5 | C6 | C7 | C8 | C9 | C10 | C11 |
|---|---|---|---|---|---|---|---|---|---|---|---|
| C1 | 0.13 | 0.13 | 0.13 | 0.13 | 0.13 | 0.13 | 0.13 | 0.13 | 0.13 | 0.13 | 0.13 |
| C2 | 0.12 | 0.12 | 0.12 | 0.12 | 0.12 | 0.12 | 0.12 | 0.12 | 0.12 | 0.12 | 0.12 |
| C3 | 0.06 | 0.06 | 0.06 | 0.06 | 0.06 | 0.06 | 0.06 | 0.06 | 0.06 | 0.06 | 0.06 |
| C4 | 0.06 | 0.06 | 0.06 | 0.06 | 0.06 | 0.06 | 0.06 | 0.06 | 0.06 | 0.06 | 0.06 |
| C5 | 0.06 | 0.06 | 0.06 | 0.06 | 0.06 | 0.06 | 0.06 | 0.06 | 0.06 | 0.06 | 0.06 |
| C6 | 0.07 | 0.07 | 0.07 | 0.07 | 0.07 | 0.07 | 0.07 | 0.07 | 0.07 | 0.07 | 0.07 |
| C7 | 0.12 | 0.12 | 0.12 | 0.12 | 0.12 | 0.12 | 0.12 | 0.12 | 0.12 | 0.12 | 0.12 |
| C8 | 0.12 | 0.12 | 0.12 | 0.12 | 0.12 | 0.12 | 0.12 | 0.12 | 0.12 | 0.12 | 0.12 |
| C9 | 0.08 | 0.08 | 0.08 | 0.08 | 0.08 | 0.08 | 0.08 | 0.08 | 0.08 | 0.08 | 0.08 |
| C10 | 0.09 | 0.09 | 0.09 | 0.09 | 0.09 | 0.09 | 0.08 | 0.08 | 0.09 | 0.09 | 0.09 |
| C11 | 0.08 | 0.08 | 0.08 | 0.08 | 0.08 | 0.08 | 0.08 | 0.08 | 0.08 | 0.08 | 0.08 |

*4.3. Using the COPRAS Method to Select the Best Alternative Strategy*

Based on the results obtained from the DANP, the dimensions considered important in adopting E-commerce in retail SMEs are technology and decision-makers. This is shown in Table 14 where these two are ranked first and second. In addition, by looking at the INRM in Figure 2, these two have positive $r - s$ values and are located at the top of the graph, meaning that they affect other dimensions. From the INRM of the decision maker's dimension shown in Figure 3, the decision maker's IT knowledge criteria (C2) has the highest $r - s$ value. This implies that it affects the other criteria in the decision-makers' dimension. Meanwhile, from the INRM of the technology dimension in Figure 3, the complexity criteria (C5) has the highest r-s value, indicating that it influences others. Based on the results obtained from the DANP, increasing the IT knowledge of decision-makers and reducing complexity in E-commerce is the basis for forming strategic recommendations relating to its adoption by retail SMEs. In accordance with the global weight of the criteria, decision-makers Innovativeness (C1), shown in Table 14, was ranked first, and this also serves as the basis for consideration. Referring to these results, four alternative strategies were recommended, as shown in Table 15.

**Table 14.** Local and global weights of dimension and criteria.

| Dimension | Weight | Criteria | Local Weight | Rank | Global Weight | Rank |
|---|---|---|---|---|---|---|
| Decision-makers (D1) | 0.253 | Decision Maker's s Innovativeness (C1) | 0.513 | 1 | 0.1300 | 1 |
| | | Decision Maker's IT Knowledge (C2) | 0.487 | 2 | 0.1232 | 2 |
| Technology (D2) | 0.250 | Relative Advantage (C3) | 0.246 | 3 | 0.0615 | 10 |
| | | Compatibility (C4) | 0.255 | 2 | 0.0639 | 9 |
| | | Complexity (C5) | 0.235 | 4 | 0.0590 | 11 |
| | | Security (C6) | 0.264 | 1 | 0.0660 | 8 |
| Organization (D3) | 0.241 | Employee's IT Knowledge (C7) | 0.494 | 2 | 0.1188 | 4 |
| | | Organization Readiness (C8) | 0.506 | 1 | 0.1218 | 3 |
| Environment (D4) | 0.246 | Customer Pressure (C9) | 0.331 | 2 | 0.0815 | 6 |
| | | Competitive Pressure (C10) | 0.348 | 1 | 0.0857 | 5 |
| | | Government Support (C11) | 0.321 | 3 | 0.0790 | 7 |

**Table 15.** An alternative strategy for E-commerce adoption in SMEs.

| Alternative Strategy | Description | References |
|---|---|---|
| IT and E-commerce skills development or training (A1) | Programs that are used both formally and informally to educate business owners about E-commerce and IT as well as to prepare them to participate in the online market | Alyoubi [65]; Amornkitvikai et al. [31]; Chau et al. [38]; Walker et al. [66] |
| Business consulting services (A2) | Support government agencies to provide a central network that offers business advice on the relevance of E-commerce in SMEs | Simpson and Docherty [67] |
| Database with previous successful experiences of managers or owners of SMEs (A3) | A database that highlights the advantages and successes of businesses that have embraced E-commerce to serve as role models for late or non-E-commerce adopters | Grandón and Ramírez-Correa [68] |
| E-commerce trial or sample software (A4) | Provide examples of E-commerce software for potential users to try, thereby motivating the use of this technology because it helps the parties involved to understand how the application works as well as aids them in gaining experience. | AlGhamdi et al. [69]; Awiagah et al. [70] |

All alternative strategies were considered feasible by each expert that used the Delphi method. Since a consensus was reached by each of them, the questionnaire was not revised, and each alternative strategy was further investigated in this study. All the experts evaluated the possible strategy based on each criterion, as shown in Table 16 using Equation (13).

**Table 16.** Initial decision-making matrix.

| Alternative Strategy | C1 | C2 | C3 | C4 | C5 | C6 | C7 | C8 | C9 | C10 | C11 |
|---|---|---|---|---|---|---|---|---|---|---|---|
| Opt | Max | Max | Max | Max | Min | Min | Max | Max | Min | Min | Max |
| Weights | 0.130 | 0.123 | 0.062 | 0.064 | 0.059 | 0.066 | 0.119 | 0.122 | 0.081 | 0.086 | 0.079 |
| A1 | 4.0 | 4.4 | 4.0 | 4.0 | 3.6 | 3.6 | 5.0 | 4.6 | 3.6 | 3.6 | 3.6 |
| A2 | 4.2 | 4.0 | 4.2 | 4.4 | 4.0 | 3.4 | 3.0 | 4.0 | 3.6 | 3.6 | 3.8 |
| A3 | 4.2 | 4.6 | 4.4 | 3.8 | 4.4 | 4.0 | 3.6 | 4.0 | 4.2 | 3.4 | 4.0 |
| A4 | 3.4 | 3.6 | 2.8 | 3.6 | 3.4 | 3.2 | 3.8 | 3.4 | 3.2 | 3.4 | 2.8 |

Equation (14) is then applied to the initial decision-making matrix to normalize it. Afterwards, the outcome was multiplied by each criterion weight using Equations (16) and (17) to create the weighted normalized decision-making matrix, as shown in Table 17.

**Table 17.** Weighted normalized decision-making matrix.

| Alternative Strategy | C1 | C2 | C3 | C4 | C5 | C6 | C7 | C8 | C9 | C10 | C11 |
|---|---|---|---|---|---|---|---|---|---|---|---|
| Opt | Max | Max | Max | Max | Min | Min | Max | Max | Min | Min | Max |
| Weights | 0.130 | 0.123 | 0.062 | 0.064 | 0.059 | 0.066 | 0.119 | 0.122 | 0.081 | 0.086 | 0.079 |
| A1 | 0.033 | 0.033 | 0.016 | 0.016 | 0.014 | 0.017 | 0.039 | 0.035 | 0.020 | 0.022 | 0.020 |
| A2 | 0.035 | 0.030 | 0.017 | 0.018 | 0.015 | 0.016 | 0.023 | 0.030 | 0.020 | 0.022 | 0.021 |
| A3 | 0.035 | 0.034 | 0.018 | 0.015 | 0.017 | 0.019 | 0.028 | 0.030 | 0.023 | 0.021 | 0.022 |
| A4 | 0.028 | 0.027 | 0.011 | 0.015 | 0.013 | 0.015 | 0.029 | 0.026 | 0.018 | 0.021 | 0.016 |

Using Equations (18)–(22), each alternative's final results and ranking were calculated as shown in Table 18. The order is determined from the highest to the least utility ($N_i$) value: A1 > A3 > A2 > A4. Based on the $N_i$ value, A1 was selected as the best alternative strategy to increase E-commerce adoption among SMEs.

**Table 18.** Final results and rankings.

| Alternative Strategy | Pi (Benefit) | Rank | Ri (Cost) | Rank | Qi | Ni | Rank |
|---|---|---|---|---|---|---|---|
| A1 | 0.191 | 1 | 0.073 | 3 | 0.264 | 100.00% | 1 |
| A2 | 0.174 | 3 | 0.073 | 2 | 0.246 | 93.05% | 3 |
| A3 | 0.182 | 2 | 0.080 | 1 | 0.249 | 94.07% | 2 |
| A4 | 0.151 | 4 | 0.067 | 4 | 0.231 | 87.35% | 4 |

## 5. Discussion

### 5.1. DANP Results

In accordance with the results of the DANP, the most critical dimensions in the adoption of E-commerce by SMEs are the decision-makers (D1) and technology (D2). However, both dimensions have the first and second highest weights (Table 14). Its importance is also reinforced by the Influential Network Relation Map (INRM) results, where these two dimensions have a positive r-s value, as shown in Figure 4. These are also located at the top of the INRM graph, as shown in Figure 2. It simply indicates that they affect other dimensions.

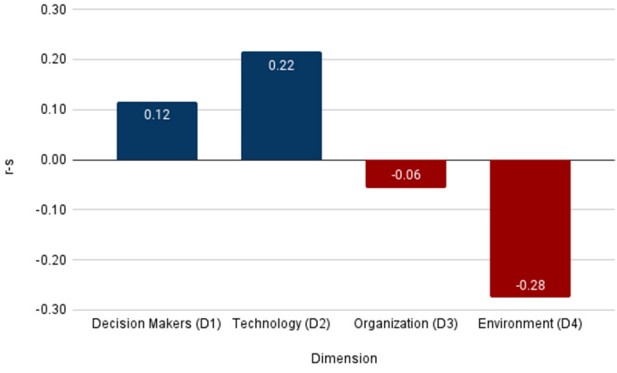

**Figure 4.** ($r - s$) value of each dimension.

Table 14 also shows that decision-makers (D1) have the largest global weight. This dimension consists of two criteria: Decision-makers' Innovativeness (C1) and Decision-makers' IT Knowledge (C2). These have the first and second-largest global weights com-

pared to the others, as shown in Table 14. In addition, the decision-maker dimension has a positive r-s value, thereby affecting the others.

The decision maker's IT knowledge (C2) is crucial in adopting E-commerce because it has the second-highest global weight after the decision maker's attitude toward innovation (C1). This is in accordance with the findings made by [44], which revealed that the owner's IT knowledge has a significant relationship with the readiness of the SME organization to adopt this technology. However, this finding contradicts the research by [18,45,71] on SMEs that E-commerce adoption is not substantially influenced by the IT knowledge of business owners and managers. On the contrary, the survey results by [18] disclosed that SMEs, which are E-commerce adopters, are owned or managed by individuals with high IT knowledge.

Based on the results of the INRM (Figure 3), it is evident that the decision makers' IT knowledge affects their Innovativeness. This implies that in adopting E-commerce technology, they need to possess adequate skills and knowledge to be innovative.

Furthermore, the technology aspect (D1) is of great concern to boost E-commerce adoption in SMEs. In addition to having the highest $r - s$ value among the other dimensions (Figure 4), it simply implies that this attribute is the most influential. This dimension also has the second largest global weight after the decision maker, as shown in Table 14. It refers to the INRM of D1 shown in Figure 3. Complexity criteria (C5) are the most influential in the technology dimension. Its importance in adopting E-commerce in SMEs aligns with the results from [11] and [72], which examined the criteria influencing this process. This contradicts the research by [29,73,74] that complexity was unimportant. Its insignificance in other studies shows that SMEs are not concerned about how E-commerce systems are operated because, over time, it becomes easier to adopt, implement, and use. There is a need to note that those studies assuming complexity are insignificant criteria are those carried out in countries such as Korea, England, and Singapore. Its importance in this study depicts that SMEs in Indonesia still perceive E-commerce as a difficult technology to understand and apply.

### 5.2. COPRAS Results

The result obtained using the COPRAS method proves that the best alternative strategy to increase E-commerce adoption for retail SMEs is developing or training IT-related skills. This strategy has a relative significance value of 0.265 and the highest utility degree at 100%. In Indonesia, the 2021 SME Onboarding Program held by Bank Indonesia (BI) in collaboration with the Indonesian E-commerce Association (idEA) was implemented to assist in adopting E-commerce for SMEs. However, it has several limitations, such as fully implemented online learning and the absence of a curriculum for E-commerce onboarding specifically for Retail SMEs. Furthermore, the onboarding materials are centered on the national marketplace and can also be used, considering that most SMEs are from the retail industry.

A blended learning program with a series of hands-on training courses is a way of adopting IT and E-commerce skills. According to all SMEs interviewed in the study by [75], its benefit is realized after completing the program. They believe it increased their e-learning skills and experiences, creating a strong foundation for sales or other long-term positive effects on the business. Furthermore, four main components of blended learning are needed in developing or training IT and E-commerce skills, as reported by [75]. This includes the existence of a supervisory team or bunch of experts, conducting offline training, online support, and platforms. Another illustration of a blended learning strategy, as indicated in the study by [76], is that it has six components: content tailored to limited time availability, self-reflection, interactivity, non-theoretical assignments or practice, case studies, and virtual networks.

According to our research results, SMEs should increase their IT competency through training programs to increase E-commerce adoption. For formal education or training, there are constraints on time and money. Expenses include the training and any revenue

lost while receiving training [77]. For this reason, the use of blended e-learning was recommended. E-learning offered advantages in the form of more flexibility and affordability. Working offline and online is advantageous since it gives participants more freedom, boosts productivity, and lowers costs.

An E-commerce adoption strategy is proposed by considering several relevant components of blended learning according to the literature study [75], with detailed strategies for each component proposed by the author considering the current E-commerce environment in Indonesia with a detailed explanation of each in Figure 5.

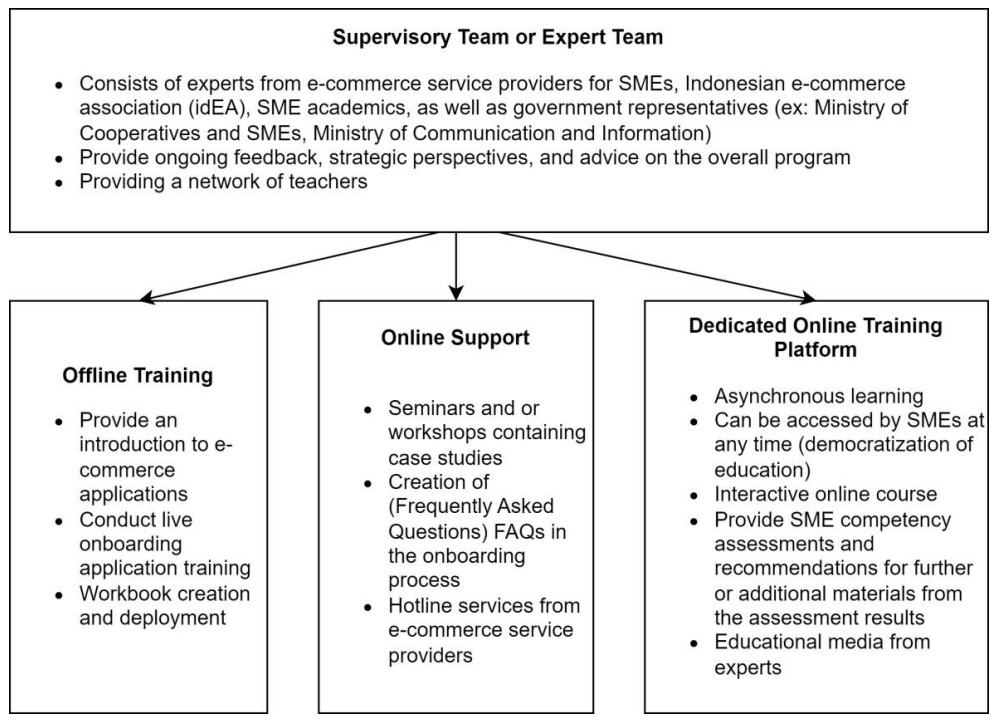

**Figure 5.** Strategy implementation design.

## 6. Conclusions

The results of this study have several implications for governmental organizations and E-commerce business providers in charge of bringing SMEs into the digital technology environment. This study utilized a hybrid MCDM framework design that involves a combination of DANP and COPRAS methods to address the dependency and feedback of each criterion. It was also used to establish the best alternative strategy to enhance the adoption of E-commerce by SMEs based on influential weights calculated using DANP. Based on the results of the analysis carried out using DANP, the decision-makers dimension (D1) has the largest influential weight, followed by technology (D2), environment (D4), and organization (D3), respectively. According to the INRM, the decision-makers (D1) and technology (D2) dimensions have positive r-s values, meaning they tend to affect others. Taking a closer look at the INRM of the technology dimension, complexity (C5) criteria has the highest $r - s$ value. Meanwhile, from the INRM of the decision-makers dimension, the decision maker's IT knowledge (C2) has the highest r-s value, implying that it affects the other criteria. Therefore, to increase E-commerce adoption by SMEs, IT development or training was selected as the best strategy with a utility degree of 100%.

This research builds upon previous studies but is different because it fills the research gap by examining the factors that are important in E-commerce adoption by Indonesian SMEs in the retail industry and their key strategy to increase utilization. This research is motivated by addressing current research gaps to evaluate the influence of criteria related to the decision-maker, technological, organizational, and environmental dimensions aligned with E-commerce adoption in the retail SME sector, propose alternative strategies to boost

E-commerce adoption and rank alternative strategies required to increase E-commerce adoption. The proposed strategy in this research can be implemented in stages with a modular program form. This is meant to divide the program into smaller interrelated modules to help SMEs better understand the information delivered. Due to the limited scope of this research on retail SMEs, further analyses need to be conducted and the results compared. There is also a need to compare the results obtained using several MCDM methodologies to rank alternative strategies to ascertain whether or not similar outcomes were realized.

**Author Contributions:** Conceptualization, M.D.; Validation, N.R.P.; Formal analysis, N.R.P.; Investigation, S.A.B.; Data curation, S.A.B.; Writing—original draft, S.A.B.; Writing—review & editing, M.D., N.R.P., J.P. and Y.C.; Supervision, M.D., J.P. and Y.C.; Funding acquisition, M.D. All authors have read and agreed to the published version of the manuscript.

**Funding:** This research was funded by Hibah PITTA Universitas Indonesia (UI) 2022.

**Institutional Review Board Statement:** Not applicable.

**Informed Consent Statement:** Informed consent was obtained from all subjects involved in the study.

**Data Availability Statement:** The data presented in this study are available on request from the corresponding author.

**Acknowledgments:** The authors are grateful to all the respondents who provided answers to the distributed online questionnaire.

**Conflicts of Interest:** The authors declare no conflict of interest.

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
