# Peer review of "E-Commerce Technologies Adoption Strategy Selection in Indonesian SMEs Using the Decision-Makers, Technological, Organizational and Environmental (DTOE) Framework"

_sustainability, doi:10.3390/su15129361_

Round 1

Reviewer 1 Report

From a general point of view, the article may be seen as interesting and well written. I do have, however, a few suggestions in order to bring some improvements of the overall quality. I will mention them in the following paragraphs, using bullets:
-    The title and the Abstract section refer to the MSMEs, while paragraph 2 (line 77), 2.1 and 2.2 speak about SMEs issues found in other literature. Same goes for Table 1, where the references are about SMEs, not MSMEs. Thus, the authors should be more precise in this regard.
-    Figure 1 is extremely small written it is almost impossible to read. Please make it larger, at least twice larger than its current size.
-    The authors should underline better both in the Abstract and Introduction sections the novelty of the conducted research in comparison with similar conducted studies.
-    Please specify whether the Figure 5 (Strategy implementation design) is taken/inspired from somewhere else or belongs 100% to the authors.
-    More details regarding the distribution of the questionnaires (what period of time, what town/city from Indonesia, how were the experts selected and why their answers are considered to be relevant to the research etc.).
-    The title says: “E-commerce Technology Adoption Strategy Design in…”, however, somehow it remains rather unclear how is this “design” part involved, from a practical perspective.
-    Regarding the criteria discussed for EC Technology Adoption, some issues worth taken into account and further added into discussion are: specialized IT Knowledge and costs involved for implementation of an EC strategy (as these were not mentioned at all, like if they would be inexistent).

Author Response

Dear Reviewers,

Thank you for all your inputs, suggestions, and questions, kindly find our answers to all inquiries in blue:

Kind Regards,

Author

______________________________________________________________________________

Reviewer 1 :

  • The title and the Abstract section refer to the MSMEs, while paragraph 2 (line 77), 2.1 and 2.2 speak about SMEs issues found in other literature. Same goes for Table 1, where the references are about SMEs, not MSMEs. Thus, the authors should be more precise in this regard
    • Have unified this as SMEs to be more precise and consistent
  • Figure 1 is extremely small written it is almost impossible to read. Please make it larger, at least twice larger than its current size.
    • Have made the diagram larger, with maximum width according to the page format provided
  • The authors should underline better both in the Abstract and Introduction sections the novelty of the conducted research in comparison with similar conducted studies.
    • Added in abstract:
      • This study aims to fill the research gap by examining the types of factors that are important in E-Commerce adoption by Indonesian SMEs in the retail industry, as well as their key strategy to increase utilization.
    • Added in introduction (with more detail):
      • This work is different from previous research in three aspects. First, despite numerous studies conducted, only a few research evaluate adoption criteria that pertain to decision-makers [17,18] but without simultaneously considering technological, organizational, and environmental factors. We evaluated all criteria using Decision-Makers, Technological, Organizational, and Environmental (DTOE) framework. Second, most literature tries to explain E-Commerce adoption patterns without suggesting ways to boost adoption [14,19–22]. Finally, although prior literature [23] offered proposals to increase E-Commerce adoption, but failed to consider which techniques should be prioritized as the primary strategy for boosting E-Commerce adoption. This research is motivated by addressing current research gaps in order to evaluate the influence of criteria related to the decision-maker, technological, organizational, and environmental dimensions aligned with E-Commerce adoption in the retail SME sector, propose alternative strategies to boost E-Commerce adoption, and rank alternative strategies required to increase E-Commerce adoption. The influence and priorities between criteria were calculated using the Decision Making Trial and Evaluation Laboratory (DEMATEL) based Analytic Network Process (ANP) method. Furthermore, an E-Commerce adoption strategy was selected using Complex Proportional Assessment (COPRAS) method.

  • Please specify whether the Figure 5 (Strategy implementation design) is taken/inspired from somewhere else or belongs 100% to the authors.
    • Explained, an E-Commerce adoption strategy is proposed by taking into account of several relevant components of blended learning according to the literature study [75], with detailed strategies for each component proposed by the author considering current e-commerce environment in Indonesiawith a detailed explanation of each in Figure 5.
  • More details regarding the distribution of the questionnaires (what period of time, what town/city from Indonesia, how were the experts selected and why their answers are considered to be relevant to the research etc.).
    • Added with below details:
    • Meanwhile, three different questionnaires were utilized for data collection. These questionnaires were distributed between January and March 2022 through an online survey from experts based in Jakarta, Indonesia. These experts were selected as they come from SME and digital economy researchers, SME e-commerce business leaders, and retail SME industry players to get a holistic response between academics, business, and industry players.
  • “E-commerce Technology Adoption Strategy Design in…”, however, somehow it remains rather unclear how is this “design” part involved, from a practical perspective.
    • To make the title less ambiguous, we have decided to change design into selection in our title. This is made because we utilized MCDM (COPRAS method) to select and rank E-Commerce strategy adoption out of few options to narrow down into one strategy and create an implementation plan to the selected strategy.
  • Regarding the criteria discussed for EC Technology Adoption, some issues worth taken into account and further added into discussion are: specialized IT Knowledge and costs involved for implementation of an EC strategy (as these were not mentioned at all, like if they would be inexistent).
    • Added additional explanation as follows, according to our research results, SMEs should increase their IT competency through training programs to increase E-Commerce adoption. For formal education or training, there are constraints on time and money. Expenses include both the training itself and any revenue lost while receiving training [77]. For this reason the use of blended e-learning was recommended. E-learning offered advantages in the form of more flexibility and affordability. Working both offline and online is advantageous since it gives participants more freedom, boosts productivity, and lowers costs.

Reviewer 2 Report

This paper uses the DTOE framework to build the evaluation index system of e-commerce technology for Indonesian MSMEs, calculates the impact and weight of each index using DEMATEL and ANP method, and then applies the COPRAS method to select the best strategy for e-commerce from the alternatives. The full text is closely combined with the reality, and has certain innovation in the method of analyzing the influence of indicators, which is completely correct in data calculation. However, the paper has the following deficiencies:

1. The method part of 3.3 is relatively confusing, mainly including formulas (4), (7), (8) and (12). Some letters need subscripts. Some subscripts or superscripts are wrong. Please check it carefully and revise it.

2. In 3.3, the letters in the method part should be unified, such as Ri and Sj, which should be lower case ri and sj.

3. Steps 6 and 7 in Section 3.4 are not clearly expressed. Step 10 can be deleted; The interpretation of Qmax in step 12 is incorrect.

4. SMES or MSMES in the text should be unified as much as possible.

5. English expression needs to be improved. For example, some of the articles use the past tense and some use the present tense, which should be carefully checked.

6. There are many problems in the format of the full text. For example, the variable is not in italics, the multiplication sign is expressed by x, and the formula (17) should be "and", not "dan".

7. COPRAS is a utility based method (Please refer to  Systems 2022, 10(5), 188; https://doi.org/10.3390/systems10050188:election of Third-Party Reverse Logistics Service Provider Based on Intuitionistic Fuzzy Multi-Criteria Decision Making), which conforms to the purpose of this paper. However, it is suggested that the author should use the data in this paper to make a comparison of methods, such as MULTIMOORA, CPT or VIKOR, to improve the credibility of the conclusion.

Author Response

Dear Reviewers,

Thank you for all your inputs, suggestions, and questions, kindly find our answers to all inquiries in blue:

Kind Regards,

Author

Reviewer 2

  • The method part of 3.3 is relatively confusing, mainly including formulas (4), (7), (8) and (12). Some letters need subscripts. Some subscripts or superscripts are wrong. Please check it carefully and revise it.
    • Have carefully revised all part 3.3 to make it easier to understand and also checked subscripts
  • In 3.3, the letters in the method part should be unified, such as Ri and Sj, which should be lower case ri and sj.
    • Have carefully revised all part 3.3 to make unified letters
  • Steps 6 and 7 in Section 3.4 are not clearly expressed. Step 10 can be deleted; The interpretation of Qmax in step 12 is incorrect.
    • Have carefully revised steps 6 and 7 part 3.4 to make it easier to understand, deleted step 10, and also checked interpretation for Qmax
  • SMES or MSMES in the text should be unified as much as possible.
    • Have unified this as SMEs to be more precise and consistent
  • English expression needs to be improved. For example, some of the articles use the past tense and some use the present tense, which should be carefully checked.
    • Have unified this as present tense to be consistent
  • There are many problems in the format of the full text. For example, the variable is not in italics, the multiplication sign is expressed by x, and the formula (17) should be "and", not "dan".
    • Have carefully revised steps 6 and 7 part 3.4 to make it easier to understand, deleted step 10, and also checked interpretation for Qmax
  • COPRAS is a utility based method (Please refer to  Systems 2022, 10(5), 188; https://doi.org/10.3390/systems10050188:election of Third-Party Reverse Logistics Service Provider Based on Intuitionistic Fuzzy Multi-Criteria Decision Making), which conforms to the purpose of this paper. However, it is suggested that the author should use the data in this paper to make a comparison of methods, such as MULTIMOORA, CPT or VIKOR, to improve the credibility of the conclusion.
    • Thank you for the input, based on our research we have selected COPRAS as our method due to following reasons: 1) it enables the simultaneous consideration of the ratio to both the ideal and negative solutions, 2) calculations are straightforward and logical, and 3) answers are obtained faster than with other approaches such as AHP and ANP [63]. The ideal solution maximizes benefits while minimizing costs. The opposite, a negative ideal solution, maximizes costs while minimizing benefits. Previous research done by Chatterjee et al. [64] discovered that the correlation coefficient values ​​between COPRAS, EVAMIX, AHP, TOPSIS, and VIKOR in the selection of materials proved that COPRAS method has the best performance. We have also acknowledged this point for future research in our conclusion as stated below:
    • Due to the limited scope of this research on retail SMEs, further analyses need to be conducted and the results compared. There is also a need to compare the results obtained using several MCDM methodologies to rank alternative strategies to ascertain whether or not similar outcomes were realized.

Reviewer 3 Report

I've put some remarks in the attachment. But, I have one huge concern regarding the research methodology:

As a research methodology, you are talking about MCDM methods and  their alternatives and criteria. But you don't have alternatives, you have dimensions. And dimensions you somehow divide into criteria. It is not good approach. The alternatives are independent of criteria. There must not be dependence between alternatives and criteria.

For instance, here is the example of something that CANNOT BE a MCDM problem:

The problem: what is the most important part of the car?

Alternatives: Engine, Tires, Seats.

Criteria: Engine power, Engine consumption, Tires performance, Seat comfortability.

This is not MCDM problem. It cannot be. Alternatives and criteria, both need to be independent. MCDM problem is something like this:

The problem: what is the best-buy car?

Alternatives: Renault Scenic, Audi Quattro, BMW X5.

Criteria: Engine power, Engine consumption, Tires performance, Seat comfortability.

So, if your dimensions cannot be independent from criteria, than you don't have a MCDM problem. But, if you believe you have MCDM problem, than make dimension independent of criteria.

Author Response

Dear Reviewers,

Thank you for all your inputs, suggestions, and questions, kindly find our answers to all inquiries in blue:

Kind Regards,

Author

Reviewer 3

I've put some remarks in the attachment.

  • Reduce the number of abbreviations in the abstract
    • Have removed all abbreviations in the abstract to be introduced later in the research
  • In the world or in Indonesia?
    • Data presented in introduction is for Indonesia, have added details on research
  • Small and Medium Enterprises should be Small and Medium-sized Enterprises
    • Have changed all ‘Medium Enterprises’ into ‘Medium-sized Enterprises’
  • Abbreviation of EC to change because can be misunderstood
    • Have changed all EC abbreviation into E-Commerce
  • Make Figure 1 larger
    • Have made the diagram larger, with maximum width according to the page format provided
  • Mentioning of other MCDM methods
    • Added with below details:
    • MCDM methods can generally be divided into two types according to their compensatory or non-compensating nature in which compensatory methods ( TOPSIS) usually combines performance which is categorized into functions to be optimized [34]. However, TOPSIS has some main drawbacks, including correlation between criteria is not considered in evaluating the Euclidean distance in TOPSIS and the ambiguity of using only objective or subjective methods to determine weights [35]. Hence this method is not suitable to analyze criteria that have interdependencies. Non-compensated methods are also known as outranking (ex. ELECTRE and PROMETHEE). Outrangking approach is used to establish a preference relation on a set of alternatives that indicates the degree of dominance among them [36]. However, an outranking approach with a non-compensating nature cannot always offer complete ranking results [37]. In the meantime, pairwise comparison in MCDM methods such as Analytic Hierarchy Proses (AHP) and Analytic Network Process (ANP) are very useful to find the weight of different criteria and compare alternatives with respect to a subjective criterion [36].
  • As a research methodology, you are talking about MCDM methods and  their alternatives and criteria. But you don't have alternatives, you have dimensions. And dimensions you somehow divide into criteria. It is not good approach. The alternatives are independent of criteria. There must not be dependence between alternatives and criteria. So, if your dimensions cannot be independent from criteria, than you don't have a MCDM problem. But, if you believe you have MCDM problem, than make dimension independent of criteria
    • Based on previous research: Chiu, W.-Y., Tzeng, G.-H., & Li, H.-L. (2013). A new hybrid MCDM model combining DANP with VIKOR to improve e-store business. Knowledge-Based Systems, 37, 48–61. doi:10.1016/j.knosys.2012.06.017, they have assessed and improve strategies to reduce the gaps in customer satisfaction caused by interdependence and feedback problems among dimensions and criteria to achieve the aspiration level, by combining the Decision Making Trial and Evaluation Laboratory (DEMATEL), DEMATEL-based Analytic Network Process (DANP), and VIšekriterijumsko KOmpromisno Rangiranje (VIKOR) methods to solve these problems. According to the function itself, DANP is an appropriate tool to include interaction and interdependence among the dimensions and criteria that appear in the cases of real world problems.

Reviewer 4 Report

E-commerce Technology Adoption Strategy Design in Indonesian MSMEs Using the Decision-Makers, Technological, Organizational, and Environmental (DTOE) Framework

l   The title needs to re-structure properly. What has been studied Introduction should be clearly stated research questions and targets first. Then answer several questions: Why is the topic important (or why do you study on it)? What are research questions or objectives? What are your contributions? Why is to propose this particular method (This must come from Literature discussion)?

l   How do you collect the questionnaire? How many experts’ responses? How are those theories related to this study ?How do you arrive Table 3 and relted to your theories  ? How is the alternatives related to your study ?

l   How are those equations related to your results ?

l   In terms of the method, for your references, Neda Attarmoghaddam, Alireza Khorakian & Amir Mohammad Fakoor Saghih (2022) Employing system dynamic and DEMATEL for improving the new product development time in knowledge-based companies, Journal of Industrial and Production Engineering, 39:7, 521-534, DOI: 10.1080/21681015.2022.2059793; and Ming-Lang Tseng, Ming K. Lim, Mohd Helmi Ali, Gabriella Christianti & Patrapapar Juladacha (2022) Assessing the sustainable food system in Thailand under uncertainties: governance, distribution and storage drive technological innovation, Journal of Industrial and Production Engineering, 39:1, 1-18, DOI: 10.1080/21681015.2021.1951858

l   The readability is low and the scientific contribution is short discussed. For your references, the three main criteria for the manuscript are: (a) quality and content of the research/review; (b) Quality, brevity and clarity of presentation; (c) Significance, relevance and timeliness of the topic. Yet, this title is (i) coverage of the literature/significant developments in the field, or clarity of discussion within an emerging topic; (ii) originality, new perspectives or insights; (iii) international interest; and (iv) relevance for governance, policy or practical perspectives relevant to the focus of this manuscript. The discussion is very weak now.

l   Please make sure your conclusions' section underscore the scientific value added of your paper, and/or the applicability of your findings/results, as indicated previously. Basically, you should enhance your findings, limitations, underscore the scientific value added of your paper, and/or the applicability of your contributions/shortages and future study in this session.

Author Response

Dear Reviewers,

Thank you for all your inputs, suggestions, and questions, kindly find our answers to all inquiries in blue:

Kind Regards,

Author

Reviewer 4

  • The title needs to re-structure properly. What has been studied Introduction should be clearly stated research questions and targets first. Then answer several questions: Why is the topic important (or why do you study on it)? What are research questions or objectives? What are your contributions? Why is to propose this particular method?
    • Answered all questions above in research with details:
      • Several recent studies focus on how e-commerce (E-Commerce) is generally adopted throughout diverse sectors. As a result, there is only a little research on its utilization in certain industries, such as retail companies [14]. In Indonesia, the distribution of SMEs is dominated by the retail, repair, and vehicle maintenance industries which constitute 46% of all businesses in the country [15]. The retail company is also the second-largest contributor to the GDP, after the processing industry [16]. As one of the largest business sectors that constitute SMEs, it played a vital role in its recovery due to the pandemic and in driving the national economy. There is a need to further investigation to ascertain how SMEs are responding to E-Commerce adoption to maintain a competitive edge during the digitization era. Due several reasons namely, limited research surrounding E-Commerce adoption in retail industries, retail being the biggest segment in Indonesian SMEs, low adoption rate of E-Commerce in Indonesian SMEs, and government support and target in digital transformation of SMEs, and, this research is conducted to answer several questions:
        • Understanding and ranking the factors that influence the decisions of Indonesian retail SMEs in the use of E-Commerce technology
        • Develop strategic recommendations to increase E-Commerce technology adoption among SMEs, especially the retail industry
      • This work is different from previous research in three aspects. First, despite numerous studies conducted, only a few research evaluate adoption criteria that pertain to decision-makers [17,18] but without simultaneously considering technological, organizational, and environmental factors. We evaluated all criteria using Decision-Makers, Technological, Organizational, and Environmental (DTOE) framework. Second, most literature tries to explain E-Commerce adoption patterns without suggesting ways to boost adoption [14,19–22]. Finally, although prior literature [23] offered proposals to increase E-Commerce adoption, but failed to consider which techniques should be prioritized as the primary strategy for boosting E-Commerce adoption. This research is motivated by addressing current research gaps in order to evaluate the influence of criteria related to the decision-maker, technological, organizational, and environmental dimensions aligned with E-Commerce adoption in the retail SME sector, propose alternative strategies to boost E-Commerce adoption, and rank alternative strategies required to increase E-Commerce adoption. The influence and priorities between criteria were calculated using the Decision Making Trial and Evaluation Laboratory (DEMATEL) based Analytic Network Process (ANP) method. Furthermore, an E-Commerce adoption strategy was selected using Complex Proportional Assessment (COPRAS) method
    • How do you collect the questionnaire? How many experts’ responses? How are those theories related to this study ?How do you arrive Table 3 and relted to your theories ? How is the alternatives related to your study ?
      • Answered all questions above in research with details:
      • How do you collect questionnaire? Meanwhile, three different questionnaires were utilized for data collection. These questionnaires were distributed between January and March 2022 through an online survey from experts based in Jakarta, Indonesia. These experts were selected as they come from SME and digital economy researchers, SME e-commerce business leaders, and retail SME industry players to get a holistic response between academics, business, and industry players.
      • How many expert response? Line 212, 220, and 227
      • How are those theories related to this study ?How do you arrive Table 3 and relted to your theories ? How is the alternatives related to your study ? These dimensions are based on the extension of the TOE theory, in which the CEO or manager makes most of the crucial choices in SMEs due to their highly centered organizational structure hence DTOE framework was applied. To support this framework, the Diffusion of Innovation (DOI) theory to analyze how, when, and to what extent the people and the business sector accept new ideas and technology that include relative advantage, complexity, and compatibility is integrated to this research.
    • How are those equations related to your results ?
      • Have added in and linked all equations to our results (ex. line 451, 455, etc)
    • Please make sure your conclusions' section underscore the scientific value added of your paper, and/or the applicability of your findings/results, as indicated previously. Basically, you should enhance your findings, limitations, underscore the scientific value added of your paper, and/or the applicability of your contributions/shortages and future study in this session.
      • Thank you for the input, we have restated and highlighted the scientific value added of our paper: This research builds upon previous study but is different because it fills the research gap by examining the types of factors that are important in e-commerce adoption by Indonesian SMEs in the retail industry, as well as their key strategy to increase utilization. This research is motivated by addressing current research gaps in order to evaluate the influence of criteria related to the decision-maker, technological, organizational, and environmental dimensions aligned with E-Commerce adoption in the retail SME sector, propose alternative strategies to boost E-Commerce adoption, and rank alternative strategies required to increase E-Commerce adoption
      • and/or the applicability of our findings/results in our conclusion: The proposed strategy in this research can be implemented in stages with a modular program form. This is meant to divide the program into smaller interrelated modules to help SMEs better understand the information being delivered. Due to the limited scope of this research on retail SMEs, further analyses need to be conducted and the results compared. There is also a need to compare the results obtained using several MCDM methodologies to rank alternative strategies to ascertain whether or not similar outcomes were realized.

Round 2

Reviewer 1 Report

The paper has been overall improved. However, the new title is still not convincing enough. There isn't only one "e-commerce technology", but many. Thus, the title should be "E-commerce Technologies Adoption...".
Also, the new title makes a reference on "selection", but this issue cannot be found at all in the paper. Therefore, I strongly advise the authors to rewrite their title in order to be a better fit for the content.

Author Response

The paper has been overall improved. However, the new title is still not convincing enough. There isn't only one "e-commerce technology", but many. Thus, the title should be "E-commerce Technologies Adoption...".
Also, the new title makes a reference on "selection", but this issue cannot be found at all in the paper. Therefore, I strongly advise the authors to rewrite their title in order to be a better fit for the content.

Thank you for the input, we have revised the title from technology to technologies, new title reference selection because we use the COPRAS method to select the best E-commerce adoption strategy out of four alternatives (Table 15) into one best strategy as seen in line 501-506, as seen in the snippet below:

Using Eq.18-23 (18) to (23), each alternative's final results and ranking were calculated as shown in table 18. The order is determined from highest to the least utility (Ni) value: A1 > A3 > A2 > A4. Based on the Ni value, A1 was selected as the best alternative

After selecting the best strategy we further discuss it in line 552-588

Reviewer 2 Report

The authors carefully revised the paper according to the comments, and the revised paper met the requirements for publication in terms of formula, language and innovation. Thank the authors for their work.

Author Response

Thank you Prof.

Reviewer 3 Report

Authors have improved their paper according to my remarks and suggestions. 

However, I still have concern about the DANP methodology. The authors have supported the usage of the methodology with the references, but my suggestion for the authors is to consider future usage of this methodology, or do not use it in the context of MCDM.

I'm MCDM expert and I'm quite sure that alternatives must be independent of criteria, and vice versa. Othervise, it is not a MCDM. I don't say that DANP methodology is wrong, I'm just saying it is wrong to call it a MCDM method. And it is my final remark, do not say that you are using MCDM method, because you are not.

I assume that DANP is using the mathematics of ANP, but it is not an ANP, therefore it is not MCDM.

I will say once again:

Here is the example of something that CANNOT BE a MCDM problem:

The problem: what is the most important part of the car?

Alternatives: Engine, Tires, Seats.

Criteria: Engine power, Engine consumption, Tires performance, Seat comfortability.

This is not MCDM problem. It cannot be. Alternatives and criteria, both need to be independent. MCDM problem is something like this:

The problem: what is the best-buy car?

Alternatives: Renault Scenic, Audi Quattro, BMW X5.

Criteria: Engine power, Engine consumption, Tires performance, Seat comfortability.

Author Response

Authors have improved their paper according to my remarks and suggestions. 

However, I still have concern about the DANP methodology. The authors have supported the usage of the methodology with the references, but my suggestion for the authors is to consider future usage of this methodology, or do not use it in the context of MCDM.

I'm MCDM expert and I'm quite sure that alternatives must be independent of criteria, and vice versa. Othervise, it is not a MCDM. I don't say that DANP methodology is wrong, I'm just saying it is wrong to call it a MCDM method. And it is my final remark, do not say that you are using MCDM method, because you are not.

I assume that DANP is using the mathematics of ANP, but it is not an ANP, therefore it is not MCDM.

I will say once again:

Here is the example of something that CANNOT BE a MCDM problem:

The problem: what is the most important part of the car?

Alternatives: Engine, Tires, Seats.

Criteria: Engine power, Engine consumption, Tires performance, Seat comfortability.

This is not MCDM problem. It cannot be. Alternatives and criteria, both need to be independent. MCDM problem is something like this:

The problem: what is the best-buy car?

Alternatives: Renault Scenic, Audi Quattro, BMW X5.

Criteria: Engine power, Engine consumption, Tires performance, Seat comfortability.

Thank you for the input, we have carefully reviewed again our study and made sure to not capture DANP as MCDM method. Aside from DANP, we have also used COPRAS as one of our methods to select the best alternative strategy to increase E-Commerce adoption. The  alternatives presented in our COPRAS method are independent of the criteria.

Round 3

Reviewer 1 Report

The paper has been improved, and it can now be accepted for publishing. However, just a few minor issues are worth mentioning:

- Figure 1 (line 192) is still unreadable, the text is too small. Perhaps redrawing it on vertical position (portrait, instead of landscape) will solve this problem.

- On the upper left part of the Figure 1, on the upper right and in the lower right squares, the word MSMEs appears again (three times), while in the text everywhere these have been replaced with SMEs.

- Figure 5 (line 564) is again referring to MSMEs, instead of SMEs (this occurs five times).

Author Response

Thank you Professor, I have corrected the paper

Reviewer 3 Report

As I said before, I still have concerns calling this MCDM, because it has completely different propositions than MCDM. 

But I will let the scientific community to judge it. 

The authors improved their paper, it can be accepted.

Author Response

Thank you Professor, I have corrected the paper. English language improvement is in progress through the MDPI service

Round 4

Reviewer 1 Report

Figure 1 and Figure 3 still need to be bigger as they are hard to read. Except these issues, the paper has been improved and can be accepted for publishing.